# Integrating whole genome and transcriptome sequencing to characterize the genetic architecture of isoform variation

Chunyu Liu [1,2,20] ✉, Roby Joehanes [3,20] ✉, Jiantao Ma[4,20], Jiuyong Xie[5], Jian Yang [1], Mengyao Wang [1], Tianxiao Huan [3], Shih-Jen Hwang [3], Jia Wen [6], Quan Sun [7,8,9], Cumhur Y. Demirkale[10], Nancy L. Heard-Costa [2,11], Peter Orchard[12], April P. Carson [13], Jeffrey W. Haessler[14], Laura M. Raffield [6], Alex P. Reiner [14,15], Nora Franceschini [6,16], Paul L. Auer [17], Charles Kooperberg [14], Yun Li [7], George O'Connor[2,18], Joanne M. Murabito [2,19], Peter Munson[3] & Daniel Levy[2,3] ✉

We present a whole-blood isoform ratio QTL (irQTL) resource by analyzing genome-wide isoform-to-gene expression ratios using sequencing data. In Framingham Heart Study (FHS, $n = 2622$) discovery, we identify over 1.1 million *cis*-irQTLs (minor allele frequency [MAF] $\geq 0.01$, $\pm 1$ Mb of 10,883 isoform transcripts, $P < 5 \times 10^{-8}$) across 4,971 genes. Among 11,425 sentinel *cis*-irQTLs, 72% replicate ($P < 1 \times 10^{-4}$) in the Women's Health Initiative (WHI; $n = 2005$). Notably, 20% of *cis*-irQTLs have no significant association with overall gene expression, indicating isoform-specific regulation. These variants are enriched at splice donor/acceptor sites and genome-wide association study loci ($P < 1 \times 10^{-10}$). We also identify 1870 sentinel *trans*-irQTLs (MAF $\geq 0.01$, $P < 1.5 \times 10^{-13}$) for 1,084 isoforms across 590 genes, and 2327 rare *cis*-irQTLs ($0.003 < MAF < 0.01$) for 2467 isoforms of 1428 genes in FHS, with external replication rates of 61% and 41% in WHI, respectively. We highlight rs12898397 in *ULK3*, which alters splice site usage and reduces expression of a full-length isoform. Mendelian randomization supports a causal role between this isoform shift and reduced diastolic blood pressure. These findings highlight the power of irQTL mapping to uncover transcript-specific regulatory mechanisms underlying complex traits.

Isoform variation results from several mechanisms, including alternative splicing, alternative transcription start sites, and alternative polyadenylation (polyA) sites[1–3]. These mechanisms generate distinct messenger RNAs (mRNAs) from the same gene, expanding proteomic diversity beyond the ~20,000 protein-coding genes in the human genome[4]. Alternative splicing, which joins exons in different combinations to produce mature RNA isoforms[5], can be both a cause and consequence of disease[6,7]. Recent advances in sequencing technologies have revealed that up to 95% of multi-exon genes undergo alternative splicing in humans[4].

While it is a dynamic process, isoform variation is under tight regulation. Many factors, including *cis*- and *trans*-regulatory elements and DNA methylation patterns, regulate isoform variation[8–10]. Several previous studies have provided crucial insights into the genetic regulation of transcriptomic isoform variation, contributing to our understanding of the genetic architecture underlying complex

diseases[11–21]. Two studies from the Genotype-Tissue Expression (GTEx) consortium used whole-genome sequencing (WGS) paired with whole-transcriptome sequencing (RNA-seq) to identify single-nucleotide polymorphisms (SNPs) affecting splice variation throughout the human genome across multiple tissues[17]. These studies established the widespread influence of genetic variation on transcriptional regulation, including splice variation. In addition to SNPs that influence the overall gene expression levels alone (i.e., expression quantitative trait loci or eQTLs), SNPs that influence splicing (with or without effects on overall gene expression) may exert a pronounced impact on complex traits[17]. The replication rate for splice variant-associated SNPs identified in these two studies, however, was low, possibly reflecting methodological differences in isoform characterization and sample size limitations[17].

Comprehensive analyses of SNP associations with isoform variation in whole blood and their relevance to human health remain limited. In prior work, we identified exon-level expression variants in 2650 genes using imputed SNPs and hybridization-based exon array data from 5257 Framingham Heart Study (FHS) participants[21]. Here, we present a new study leveraging WGS and RNA-seq data from 3716 FHS participants (2622 for discovery, 1094 for internal replication) to map isoform ratio QTLs (irQTLs) as described originally by Yamaguchi et al.[18] (Supplementary Data 1). We further evaluate these findings in two external cohorts with WGS and RNA-seq data[22,23]. Beyond identifying irQTLs, we distinguish those with little or no impact on gene-level expression, assess overlap with genome-wide association study (GWAS) loci[24,25], and use Mendelian randomization (MR)[26] to evaluate potential causal effects on cardiovascular disease (CVD) risk factors (Fig. 1). We highlight three irQTL-transcript pairs to illustrate how this resource can reveal candidate causal variants influencing disease through transcriptomic regulation.

## Results

We analyzed 12,887,093 SNPs (Hardy–Weinberg equilibrium, [HWE] $P \geq 1 \times 10^{-10}$) and isoform ratios of 25,642 transcripts (Trimmed Mean of M-values [TMM] $\geq 2$ and <20% missing values) from 10,150 genes having more than one transcript in the FHS discovery sample ($n = 2622$) (see "Methods" section) to identify irQTLs, which capture how genetic variants influence the relative abundance of different transcript isoforms from the same gene[18] (see "Methods" section). Replication was performed internally in an additional FHS sample ($n = 1094$) and externally in two cohorts: WHI[23] ($n = 2005$), using WGS and whole-blood RNA-seq data, and the JHS[22] ($n = 1010$), using WGS and RNA-seq from peripheral blood mononuclear cells (PBMCs) (Table 1, Supplementary Data 2).

### Discovery of transcriptome-wide common *cis*-irQTLs

A *cis*-irQTL was defined as a variant within 1 Mb of the isoform's transcription start site (TSS). We identified over 1.1 million common *cis*-irQTLs (minor allele frequency [MAF] $\geq 0.01$) associated with 10,883 isoform transcripts, representing more than 3.5 million significant *cis*-irQTL-isoform pairs ($P < 5 \times 10^{-8}$) across 4971 genes. Of these, we prioritized 11,425 sentinel variants forming 14,056 lead pairs with the same isoforms (Supplementary Data 3). A sentinel *cis*-irQTL is the variant most strongly (i.e., with the lowest *p*-value) associated with an isoform. An isoform may be associated with multiple sentinel *cis*-irQTLs if they are in mutual perfect linkage disequilibrium (LD $r^2 = 1$). The top *cis*-irQTL-isoform pairs are listed in Table 2. The most significant sentinel *cis*-irQTL, rs4841, was positively associated with the

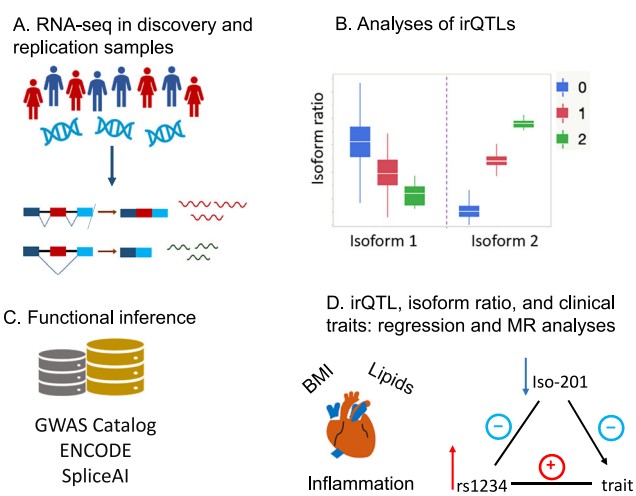

**Fig. 1 | Schematic outline. A** RNA sequencing was performed on whole-blood samples from the Framingham Heart Study (FHS, discovery cohort) and the Women's Health Initiative (WHI, replication cohort). These stored samples from community-based populations enabled genome-wide analysis of both sequence and isoform-specific expression across thousands of genes. **B** Expression quantitative trait loci (eQTL) and isoform ratio QTL (irQTL) analyses were performed to identify genetic variants associated with overall gene expression or the relative abundance of specific transcript isoforms. Variant allele dosage (0, 1, or 2 copies) may correspond to low, intermediate, or high expression levels of particular isoforms, reflecting distinct regulatory patterns. Box plots illustrate the distribution of expression across genotype groups, showing the median (central line), interquartile range (box), and range of non-outlier values (whiskers), with individual outliers plotted beyond 1.5 × interquartile range (IQR). This panel is schematic and intended for illustrative purposes only; no statistical tests were performed. **C** Functional consequences of genetically driven isoform regulation were evaluated using tools such as Gene Set Enrichment Analysis and external resources, including the GWAS Catalog, ENCODE, and SpliceAI, which predicts the impact of genetic variants on RNA splicing. **D** The clinical relevance of irQTLs was assessed by integrating GWAS data and applying Mendelian randomization to identify causal links between genetic variants, alternative splicing, and cardiovascular or hematological traits. This framework connects genotype to splicing variation and ultimately to disease risk or trait expression.

**Table 1 | Participant characteristics of the discovery and replication samples in the Framingham Heart Study (FHS) and Women's Health Initiative (WHI)**

| Variable mean (SD) or % | FHS | | WHI external replication (n = 2005) |
| --- | --- | --- | --- |
| | Discovery (n = 2622) | Internal replication (n = 1094) | |
| Women | 53.7 | 56.4 | 100 |
| Age, years | 52.9 (14.0) | 52.1 (12.7) | 79.2 (6.7) |
| BMI, kg/m2 | 28.0 (5.7) | 28.0 (5.8) | 28.7 (6.0) |
| SBP, mmHg | 118.6 (15.6) | 119.0 (15.0) | 126.9 (15.1) |
| DBP, mmHg | 73.4 (9.6) | 73.8 (9.2) | 72.7 (9.1) |
| Fasting glucose, mg/dL | 98.6 (20.8) | 97.1 (15.3) | 100.4 (30.4) |
| TC, mg/dL | 185.9 (35.9) | 188.9 (37.0) | 195.7 (39.6) |
| HDL, mg/dL | 60.1 (18.2) | 60.9 (18.5) | 60.4 (15.0) |
| Trig, mg/dL | 114.2 (77.9) | 114.6 (74.4) | 104.9 (52.4) |
| LDL, mg/dL | 103.1 (30.8) | 105.1 (31.1) | 114.4 (35.0) |
| Current smoking | 8.6 | 10.7 | 2.6 |
| Hypertension | 64.3 | 49.4 | 82.1 |
| Diabetes | 7.5 | 5.7 | 34.3 |
| HRX | 26.7 | 24.9 | 42.6 |
| LIPIDRX | 24.8 | 23.0 | 40.3 |
| DMRX | 5.7 | 4.0 | 14.1 |

*BMI* body mass index, *SBP/DBP* systolic/diastolic blood pressure, *TC* total cholesterol, *HDL* high-density lipoprotein, *Trig* triglyceride, *LDL* low-density lipoprotein, *HRX* treatment for hypertension, *LIPIDRX* treatment for high lipid level, *DMRX* treatment for diabetes.

**Table 2 | Top cis-irQTL-isoform pairs identified in the Framingham Heart Study and replication in Women's Health Initiative**

| Sentinel SNP | A:EA | Isoform | Chr | SNP Pos | Gene Symbol | FHS (n = 2622) | | | | WHI (n = 2005) | | | |
|---|---|---|---|---|---|---|---|---|---|---|---|---|---|
| | | | | | | EAF | R² | Beta | −logP | EAF | R² | Beta | −logP |
| rs4841 | C:T | ENST00000519690.1 | 5 | 150446963 | RPS14 | 0.25 | 0.96 | 0.045 | 1859 | 0.23 | 0.90 | 0.02 | 1004 |
| rs10774671 | G:A | ENST00000202917.10 | 12 | 112919388 | OAS1 | 0.62 | 0.95 | −0.258 | 1714 | 0.55 | 0.95 | −0.26 | 1249 |
| rs73191241 | A:G | ENST00000388962.4 | 12 | 108619543 | SELPLG | 0.19 | 0.92 | 0.095 | 1443 | 0.14 | 0.77 | 0.11 | 637 |
| rs62477626 | T:C | ENST00000475668.6 | 7 | 142065610 | MGAM | 0.19 | 0.92 | −0.352 | 1402 | 0.12 | 0.72 | −0.26 | 549 |
| rs7513045 | G:T | ENST00000495526.5 | 1 | 39738494 | PPIE | 0.36 | 0.91 | 0.219 | 1314 | 0.30 | 0.88 | 0.08 | 893 |
| rs2318043 | A:G | ENST00000381605.8 | 20 | 1615827 | SIRPB1 | 0.21 | 0.9 | −0.261 | 1297 | 0.20 | 0.73 | −0.21 | 560 |
| rs41280971 | G:T | ENST00000493091.5 | 7 | 100356834 | PILRB | 0.18 | 0.9 | 0.15 | 1284 | 0.15 | 0.81 | 0.07 | 702 |
| rs2425068 | T:C | ENST00000437340.5 | 20 | 35626801 | CPNE1 | 0.08 | 0.88 | 0.152 | 1169 | 0.05 | 0.84 | 0.12 | 788 |
| rs12898397 | T:C | ENST00000440863.7 | 15 | 74837752 | ULK3 | 0.6 | 0.87 | −0.263 | 1154 | 0.39 | 0.73 | −0.21 | 561 |
| rs7309653 | A:G | ENST00000541272.1 | 12 | 124910282 | UBC | 0.79 | 0.86 | 0.012 | 1103 | 0.54 | 0.87 | 0.05 | 875 |
| rs13320 | C:T | ENST00000366814.8 | 1 | 151874041 | THEM4 | 0.58 | 0.86 | 0.165 | 1082 | 0.53 | 0.70 | 0.11 | 523 |
| rs14434 | C:T | ENST00000460851.6 | 3 | 150584223 | EIF2A | 0.86 | 0.85 | −0.291 | 1045 | 0.74 | 0.80 | −0.17 | 681 |
| rs28374262 | A:G | ENST00000465395.5 | 2 | 218234030 | ARPC2 | 0.49 | 0.85 | 0.008 | 1044 | 0.32 | 0.67 | 0.01 | 483 |
| rs2724360 | C:T | ENST00000357714.5 | 1 | 207769813 | CD46 | 0.8 | 0.84 | −0.149 | 1012 | 0.78 | 0.80 | −0.15 | 686 |
| rs2241976 | A:G | ENST00000418251.6 | 2 | 113185893 | PSD4 | 0.41 | 0.84 | 0.101 | 1011 | 0.55 | 0.73 | 0.08 | 568 |
| rs7063 | A:T | ENST00000296754.7 | 5 | 96774507 | ERAP1 | 0.3 | 0.83 | 0.165 | 1000 | 0.22 | 0.79 | 0.17 | 676 |
| rs7171507 | T:C | ENST00000564570.1 | 15 | 75444946 | MAN2C1 | 0.28 | 0.83 | 0.038 | 1000 | 0.38 | 0.80 | 0.07 | 688 |
| rs9616714 | G:A | ENST00000405854.5 | 22 | 49657211 | *C22orf34 | 0.63 | 0.83 | 0.298 | 995 | 0.45 | 0.74 | 0.29 | 571 |
| rs452717 | T:G | ENST00000493242.1 | 19 | 54277224 | LILRB2 | 0.85 | 0.83 | −0.129 | 990 | 0.73 | 0.78 | −0.10 | 645 |
| rs10493821 | C:T | ENST00000489444.6 | 1 | 89009452 | GBP3 | 0.34 | 0.82 | 0.29 | 963 | 0.21 | 0.74 | 0.25 | 571 |
| rs72494581 | T:C | ENST00000514448.5 | 5 | 181003797 | BTNL8 | 0.3 | 0.82 | 0.135 | 959 | 0.27 | 0.63 | 0.12 | 426 |
| rs372699745 | TA:T | ENST00000343139.9 | 6 | 32593216 | HLA-DQA1 | 0.29 | 0.82 | −0.208 | 951 | 0.25 | 0.56 | −0.15 | 357 |
| rs7724759 | G:A | ENST00000348386.7 | 5 | 96740783 | CAST | 0.33 | 0.82 | 0.1 | 947 | 0.22 | 0.78 | 0.09 | 658 |
| rs11382548 | C:CA | ENST00000334888.9 | 11 | 61398259 | TMEM216 | 0.85 | 0.81 | 0.147 | 938 | 0.82 | 0.58 | 0.14 | 369 |
| rs34634498 | G:A | ENST00000376228.9 | 6 | 31268623 | HLA-C | 0.11 | 0.81 | −0.081 | 924 | 0.12 | 0.58 | −0.11 | 375 |
| rs930232 | G:A | ENST00000263097.9 | 19 | 1036019 | CNN2 | 0.45 | 0.61 | −0.032 | 532 | 0.43 | 0.43 | −0.04 | 244 |
| rs930232 | G:A | ENST00000348419.7 | 19 | 1036019 | CNN2 | 0.45 | 0.64 | 0.030 | 586 | 0.43 | 0.39 | 0.03 | 213 |
| rs5014188 | T:C | ENST00000568865.3 | 19 | 1037109 | CNN2 | 0.88 | 0.58 | −0.0034 | 481 | 0.83 | 0.58 | −0.01 | 375 |

A:EA allele:effective allele, EAF effective allele frequency, $R^2$ variance in an isoform ratio that was explained by the sentinel irQTL, −logP −log10 (p-value), FHS Framingham Heart Study, WHI Women's Health Initiative. Linear regression was performed between the irQTL and genotype dosage, with significance assessed using a two-sided t-test for the null hypothesis, beta = 0. Significance was defined as −log₁₀(P) > 7.301 to account for multiple testing. A random sentinel cis-irQTL was selected per gene if more than one cis-irQTL was associated with that gene. *All genes, except for C22orf34 (a long intergenic non-coding RNA), are protein-coding genes.

A

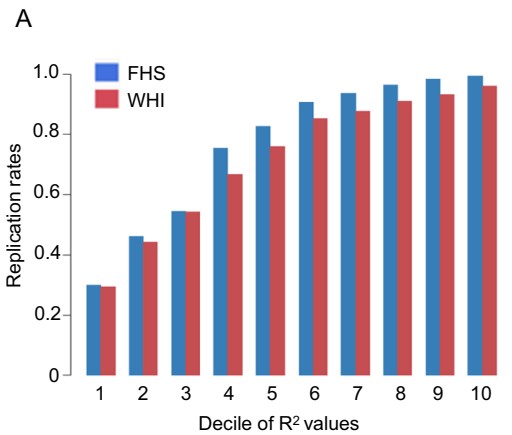

B

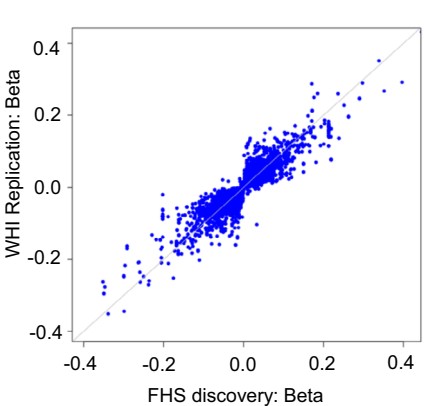

**Fig. 2 | Replication of sentinel *cis*-irQTLs in relation to their associated isoforms.** We identified 11,425 common variants (MAF ≥ 0.01) as sentinel *cis*-irQTLs for 10,883 isoforms across 4971 genes in the FHS discovery sample ($n = 2622$). Internal replication was performed in an independent FHS sample ($n = 1094$), and external replication in the WHI cohort ($n = 2005$). A sentinel *cis*-irQTL-isoform ratio association was considered replicated if its $P < 1 \times 10^{-4}$ in the replication sample and showed the same direction of effect as in the discovery sample. Replication rate was defined as the proportion of discovered pairs that met these criteria in the replication sample. **A** Replication rates (y-axis) increased with $R^2$ (x-axis), proportion of isoform ratio variance explained by the irQTL. Average replication rates were 76.7% in FHS and 72.4% in WHI. **B** Comparison of effect sizes between the FHS discovery (x-axis) and WHI replication (y-axis) samples showed strong consistency (Pearson $r = 0.91$; 95%CI: 0.90, 0.91), with 99.8% of *cis*-irQTL-isoform pairs sharing the same direction of association. MAF, minor allele frequency.

ratio of isoform ENST00000519690.1 of *RPS14,* and explained 96% of the variance ($R^2$) of the isoform ratio in the discovery sample. *RPS14* encodes ribosomal protein 14, which plays a crucial role in ribosome biogenesis and protein synthesis, as well as cell proliferation and nuclear stress[27,28].

The irQTL method, introduced by Yamaguchi et al.[18], may identify QTLs that affect splicing, as well as QTLs associated with changes in isoform ratios resulting from other regulatory mechanisms. To better interpret the molecular consequences of each sentinel variant, we integrated multiple annotation layers from the following databases: GENCODE[29], SpliceAI[30], SNPEff[31], and Homer[32]. Each database offers distinct strengths and limitations, capturing different aspects of variant annotation. SnpEff[31] is a variant annotation tool that primarily uses Ensembl-based gene models, but it also supports RefSeq, GENCODE, and UCSC Genome Browser annotations. According to SnpEff, 279 variants are related to splicing (e.g., as splice acceptor, splice donor, or splice region variants). SpliceAI[30], a tool for predicting splice site alterations from genetic variants, predicted that 215 variants had SpliceAI prediction scores >0.2, suggesting potential splice-altering effects, and 82 of these had SpliceAI prediction scores >0.5, indicating a higher likelihood of splicing disruption. A total of 86 variants were identified by both SnpEff and SpliceAI as potentially splice-altering (Supplementary Data 3).

### Discovery of transcriptome-wide common *trans*-irQTLs
A *trans*-irQTL was defined as being located more than 1 Mb away from the TSS of an isoform or on a separate chromosome. Using a significance threshold of $P < 1.5 \times 10^{-13}$, we identified over 130,000 *trans*-irQTLs associated with 1084 isoforms from 590 genes. For each isoform, we selected the most significant *trans*-irQTL per chromosome (22 autosomes and X chromosome), yielding 1870 sentinel *trans*-irQTLs in the discovery sample (Supplementary Data 4). Notably, the largest fraction of these pairs (31%) involved transcripts from genes on chromosome 6 that were significantly associated with irQTLs across the genome, with more than half of these genes located in the HLA (Human Leukocyte Antigen)[33] region. Additionally, over 75% of these pairs involved irQTLs and gene transcripts located on different chromosomes (Supplementary Fig. 1).

The most significant of these inter-chromosomal pairs was rs1458255–ENST00000418824, with the irQTL on chromosome 4 explaining 54% of the variance of the isoform ratio of *RPL9P9*

(ribosomal protein L9 pseudogene 9), located on chromosome 15. *RPL9P9* is a non-functional copy of the *RPL9* gene, and the latter encodes a ribosomal protein that is a component of the 60S ribosomal subunit[34]. Five of the sentinel variants were annotated as splicing-related by SnpEff, and one had a SpliceAI score greater than 0.2 (rs55934523 C > G variant in *FBRSL1* with a score of 0.54 for acceptor gain) (Supplementary Data 4).

### Discovery of transcriptome-wide rare *cis*-irQTLs
We assessed *cis*-irQTL variants (0.003 < MAF < 0.01) in the discovery sample. This analysis identified 2327 rare *cis*-irQTL variants associated with 2467 isoforms ($P < 5 \times 10^{-8}$), yielding 3102 significant *cis*-irQTL-isoform pairs across 1428 genes (Supplementary Data 5). Among these genes, 6% had isoforms that were not associated with common variants. Of note, some rare variants explained a larger $R^2$ of the corresponding isoform ratios than any common *cis*-irQTL variant for the same gene. For example, the rare variant rs1128271 (MAF = 0.0093) explained approximately 38% of the variance in isoform ratios of ENST00000301788.12 in *POLR2G* --much more than the strongest *cis*-irQTL, rs144760211 (MAF = 0.03), which explained just 9% of the variance in the isoform ratio (Supplementary Data 3 & 5). Twenty-eight sentinel variants were annotated as splicing-related by SnpEff, and seven of them had a SpliceAI score greater than 0.2 (Supplementary Data 5).

### Replication of common *cis*-irQTLs
An irQTL-isoform pair was considered replicated if it was found in the replication sample with a $P < 1 \times 10^{-4}$ and showed the same direction of association as in discovery. The replication rate was defined as the proportion of discovered irQTL-isoform pairs present in the replication samples that met the predefined replication criteria.

Among the 14,056 discovered sentinel *cis*-irQTL-isoform pairs ($P < 5 \times 10^{-8}$), 10,783 (76.7%) were internally replicated in an additional 1094 FHS samples, and 10,174 (72.4%) were externally replicated in the WHI (Fig. 2, Supplementary Data 6). $R^2$ represents the proportion of variance in an isoform ratio explained by its corresponding irQTL. Replication rates increased with higher $R^2$ values, as demonstrated in Fig. 2A. For example, among irQTLs in the second decile, replication rates were approximately 46% in FHS and 44% in WHI, whereas in the eighth decile, these rates increased to 98% and 93%, respectively (Fig. 2A). We also compared the effect sizes and directions of

association for the 10,194 *cis*-irQTL-isoform pairs present in both the discovery and WHI replication samples. Approximately 99.8% ($n = 10,174$) of these pairs exhibited the same direction of association and highly concordant effect sizes (Pearson coefficient $r = 0.91$) (Fig. 2B, Supplementary Data 3).

The overall replication rate of *cis*-irQTLs with MAF ≥ 0.01 in JHS ($n = 1,010$) was 32.4%, substantially lower than in WHI, though replication improved with increasing *cis*-irQTL $R^2$ values (Supplementary Data 3 & 6, Supplementary Fig. 2). JHS isolated RNA from PBMCs[22], whereas FHS and WHI used whole-blood-derived RNA, limiting direct comparability (Supplementary Data 7). To further evaluate the effects of RNA source, ancestry, and sample size, separate analyses were conducted in African American ($n = 918$) and White American ($n = 1000$) participants in WHI. Replication rates were lower in the White American sub-sample (60.2%), reflecting its smaller sample size, and even lower in the similarly sized African American sub-sample (49.0%), reflecting the potential influence of ancestry. This analysis suggests that the reduction in replication rate observed in the JHS cohort (32.4%) may largely reflect the use of a different RNA source.

A prior splicing QTL study in FHS identified exon-level QTLs in 2650 genes using imputed SNPs and exon array data[21]. That study also reported the most significant splicing QTLs within 50 kb of each gene. Only 458 (17.3%) of gene-SNP pairs from the earlier study showed $P < 1 \times 10^{-4}$ in our discovery sample (Supplementary Data 8). This modest replication rate likely reflects differences in technology (imputed SNPs vs. WGS; exon arrays vs. RNA-seq) and methodology (SNPs associated with exon expression vs. isoform ratio changes) between the earlier and current studies.

### Replication of common *trans*-irQTLs

Using the same replication criteria described earlier, the average replication rates of the 2999 sentinel *trans*-irQTL-isoform pairs were 80.8% in FHS and 60.8% in WHI. Similarly, replication rates increased with $R^2$ values of the *trans*-irQTLs (Supplementary Data 4 & 9, Supplementary Fig. 3). The replication rate of *trans*-irQTL-isoform pairs was again substantially lower in JHS (average rate = 20.2%) compared to WHI.

### Replication of rare *cis*-irQTLs

Using the same replication criteria, replication rates among the 3102 significant *cis*-irQTL-isoform pairs involving rare variants increased with higher $R^2$ values of the *cis*-irQTLs (Supplementary Data 5 & 10, Supplementary Fig. 4). However, overall replication rates of rare variant *cis*-irQTL-isoform pairs remained substantially lower than those observed for common variants, only 22.2% in the additional 1094 FHS sample and 40.7% in WHI. The lower replication rate in FHS likely reflects its smaller sample size ($n = 1094$) compared to WHI ($n = 2005$). JHS ($n = 1010$) showed a markedly lower replication rate of just 9.4%.

### Comparison of common *cis*-irQTLs and *cis*-eQTLs

We previously reported 6,778,286 significant *cis*-eQTL-gene pairs from 2,855,111 *cis*-eQTLs (MAF ≥ 0.01) associated with 15,982 genes (at $P < 5 \times 10^{-8}$), using the same sample as in the current irQTL discovery analysis ($n = 2622$)[35]. Based on *cis*-eQTL and *cis*-irQTL analyses, about 80% ($n = 3997$ of 4971) of genes with irQTLs also had eQTLs (Fig. 3A), indicating a strong overlap between splicing regulation (irQTLs) and gene expression regulation (eQTLs). In the remaining 20% of genes significantly associated with irQTLs but not eQTLs, isoform usage changes occurred independently of total expression levels, suggesting that alternative transcript usage can play a key regulatory role without significantly affecting overall gene expression.

We analyzed 4974 genes harboring both irQTLs and eQTLs, and their sentinel cis − variants (MAF ≥ 0.01) by comparing the proportion of $R^2$ for isoform ratios to that for overall gene expression (i.e., as *cis*-eQTLs). Based on this comparison, the SNPs were categorized into three groups: 1946 with stronger irQTL effects (higher $R^2$ for isoform ratio), 2716 with comparable irQTL and eQTL effects, and the rest with stronger eQTL effects (higher $R^2$ for overall gene expression). For example, rs10774671 G>A in *OAS1* was a strong *cis*-irQTL ($R^2 = 0.95$) for the isoform ratio of ENST00000202917.10, but a weak eQTL for overall *OAS1* expression ($R^2 = 0.05$) (Fig. 3B).

We compared the predicted functions of sentinel *cis*-eQTLs and *cis*-irQTLs based on their physical locations (Supplementary Data 11). Compared to eQTLs, irQTLs were more enriched in splice donor sites (median fold enrichment = 25.2 vs. 9.2; two-sided *t*-test, $P = 4.3 \times 10^{-56}$) and splice acceptor sites (median fold enrichment = 12.6 vs. 5.9, $P = 8.8 \times 10^{-127}$) (Fig. 3C). In contrast, eQTLs were more enriched in the upstream region of a gene (within 5 kb) (median fold enrichment=1.31 vs. 0.84, $P < 1 \times 10^{-300}$) or in the 5′-untranslated region (UTR, median fold enrichment = 3.0 vs. 2.5, $P < 2.7 \times 10^{-188}$) (Supplementary Data 11). Additionally, top irQTLs showed greater enrichment in RNA-binding protein (RBP) enhanced crosslinking and immunoprecipitation (eCLIP) peaks than top eQTLs (mean fold-enrichment 3.6 vs. 2.0, $P < 2 \times 10^{-16}$, see "Methods" section) (Fig. 3D).

### Enrichment of *cis*-irQTLs in GWAS Catalog SNPs

We linked *cis*-irQTLs to the GWAS Catalog SNPs to investigate whether *cis*-irQTLs are associated with diseases or traits[25]. Common *cis*-irQTL variants exhibited enrichment in GWAS Catalog variants that were associated with hundreds of traits, such as blood protein measurement (fold enrichment = 7.0, $P < 1 \times 10^{-300}$), BMI-adjusted waist-hip ratio (fold enrichment = 6.4, $P < 1 \times 10^{-300}$), and high-density lipoprotein cholesterol level (fold enrichment = 4.6, $P < 1 \times 10^{-300}$) (Supplementary Data 12). These findings were consistent with those from our prior *cis*-eQTL enrichment analysis[35], reflecting a sizable proportion of overlap of *cis*-irQTLs and *cis*-eQTLs. In addition, we identified 197 GWAS traits enriched for *trans*-irQTL variants, including traits such as BMI-adjusted waist-hip ratio (fold enrichment = 16.8, $P < 1 \times 10^{-300}$) (Supplementary Data 13).

To demonstrate the utility of this irQTL resource in understanding the role of isoform regulation, particularly splicing, in relation to complex traits and diseases, we highlight three representative examples featuring top irQTLs from our analysis.

### *OAS1* harbors a sentinel irQTL linked to isoform regulation

The rs10774671 G>A variant was the 2nd most significant lead *cis*-irQTL in the entire analysis and is associated with the expression ratio of transcript ENST00000202917.10 (*OAS1*-201) in *OAS1* at 12q24.13 (Fig. 4A and Table 2). *OAS1* produces seven detectable isoform transcripts in FHS (Supplementary Data 14). *OAS1* has long been known to play a role in the response to viral infection and diabetes risk, and has recently been implicated in COVID-19 severity[36–40]. The rs10774671-A allele disrupts the conserved dinucleotide "AG" of the splice acceptor site of intron 5, changing it to "AA" (Fig. 4B, C). This resulted in a lower isoform ratio of *OAS1*-201 ($R^2 = 0.95$, beta = −0.26, $P = 1 \times 10^{-1713}$) and higher isoform ratios of the other isoforms, including *OAS1*−203 (beta = 0.11, $P = 1 \times 10^{-1359}$) (Fig. 4D).

According to SpliceAI[30], the rs10774671-A allele has a high probability score of 0.89 for splice acceptor loss in intron 5. To directly visualize the effects of this splice acceptor loss, we analyzed RNA-seq data from a randomly selected participant for each of the three rs10774671 genotypes: G/G, A/G, and A/A. The participant with the G/G genotype (i.e., carrying the canonical 'AG' acceptor site) exhibited complete exclusion of intron 5 in the mature mRNA isoform. Conversely, the participant with the A/A genotype (i.e., with complete loss of the acceptor site) showed full retention of intron 5 in mRNA. In contrast, the participant with the A/G genotype displayed partial retention of intron 5 (Fig. 4C).

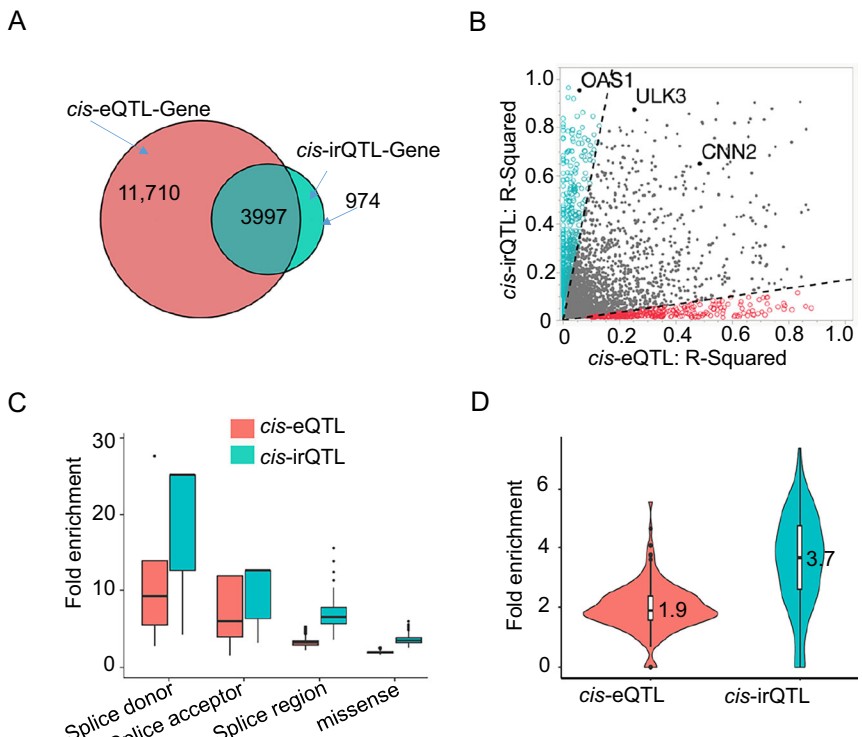

**Fig. 3 | Comparison of *cis*-irQTLs and *cis*-eQTLs. A** Among 2622 Framingham Heart Study participants, more genes had *cis*-eQTLs[35] ($n = 15,707$) than *cis*-irQTLs ($n = 4971$). **B** Association strength ($R^2$) of *cis*-irQTLs versus *cis*-eQTLs across 4971 irQTL genes in 2622 participants. Dashed lines separate SNPs into stronger irQTLs (cyan, $n = 1946$), overlapping signals (middle, $n = 2716$), and stronger eQTLs (red). Highlighted genes (*OAS1, ULK3, CNN2*) are discussed in the main text. **C** Fold enrichment of sentinel *cis*-eQTLs and *cis*-irQTLs in selected genomic features based on illustrative regions. Enrichment was calculated using 1000 matched SNP sets based on their MAFs and TSS distance. Fold enrichment reflects the observed overlap relative to the mean across matched null sets and exhibits a non-normal distribution. Box plots show the distribution of fold enrichment across replicates; boxes represent the interquartile range, with the horizontal line indicating the median. Based on median (IQR; range), irQTLs showed greater enrichment than eQTLs at splice donor (25.2 [12.6–25.2; 4.2–25.2] versus 9.2 [5.5–13.8; 2.8–27.6];

$P = 4 \times 10^{-56}$), splice acceptor (12.6 [6.3–12.6; 3.2–12.6] versus 5.9 [4.0–11.9; 1.5–11.9]); $P = 9 \times 10^{-127}$, splice region (6.5 [5.6–7.7; 3.5–15.5] versus 3.2 [2.9–3.4; 2.2–5.2]; $P < 2.2 \times 10^{-308}$), and missense sites (3.5 [3.2–3.8; 2.5–5.9] versus 1.9 [1.8–2.0; 1.6–2.6]; $P < 2.2 \times 10^{-308}$). For splice donor and acceptor sites, the median equals the maximum, indicating uniformly strong enrichment. Two-sided Wilcoxon rank-sum tests compared fold enrichment distributions between irQTLs and eQTLs within genomic regions. **D** Fold enrichment of sentinel *cis*-eQTLs and *cis*-irQTLs in RNA-binding protein (RBP) binding sites from 203 in eCLIP datasets profiled by ENCODE, showing stronger enrichment of irQTLs in functional RBP regions. Fold enrichment analysis was performed using the same method described above. Median enrichment (IQR; range) was 1.9 (1.6, 2.4; 0, 5.7) for *cis*-eQTLs and 3.7 (2.6, 4.8; 0, 7.4) for *cis*-irQTLs. MAF minor allele frequency, HWE Hardy–Weinberg Equilibrium, IQR interquartile range, TSS transcript start site.

In addition to its well-known role in impacting COVID-19 severity, the rs10774671-A variant was reported to be significantly associated with lower WBC count (beta = −0.013, $P = 3.2 \times 10^{-11}$) in a large GWAS[41] (Fig. 4E). Of note, our MR analysis suggested that a lower isoform ratio of *OAS1*-201 was putatively causal for a lower WBC count (MR beta = 0.053, $P < 6.5 \times 10^{-10}$) (Fig. 4E, Supplementary Data 15). These findings support a causal role for this variant in COVID-19 severity.

**An irQTL in *ULK3* alters isoform usage and protein structure**
The *ULK3* gene (unc-51-like kinase 3, at 15q21.1) is a serine/threonine protein kinase gene on the negative strand of chromosome 15 (15q21.1), involved in regulating sonic hedgehog signaling and autophagy[42,43]. It contains 16 exons and encodes 11 detectable isoform transcripts in FHS (Supplementary Data 14). Isoform ENST00000440863.7 (*ULK3*-201) produces a full-length 472-amino acid protein, whereas ENST00000569437.5 (*ULK3*-220) produces a truncated 470-amino acids isoform, missing two amino acids (valine and lysine, "VK") at the 5' of the 14th exon (Fig. 5A, Supplementary Fig. 5). The sentinel *cis*-irQTL rs12898397 (T>C), located in exon 14, was strongly associated with lower expression ratio of *ULK3*-201 ($R^2 = 0.87$, $\beta = −0.26$, $P = 1 \times 10^{-1154}$) (Table 2) and higher expression ratio of *ULK3*-220 ($R^2 = 0.63$, $\beta = 0.16$, $P = 1 \times 10^{-573}$) (Fig. 5B). Since *ULK3* is transcribed from the negative strand, we converted the sequence surrounding

rs12898397 to its complementary form (e.g., T>C becomes A>G at rs12898397), while preserving the 5' to 3' orientation for downstream analyses.

We sought to understand the phenotypic consequences of these isoform proportion changes. The rs12898397-G allele, previously associated with lower diastolic blood pressure (DBP) in a GWAS[44], was also associated with a reduced expression ratio of the *ULK3*-201 isoform. Using rs12898397 as the genetic instrument in MR, we found that a lower expression ratio of *ULK3*-201 was causally associated with lower DBP (MR beta = 1.24, $P = 1.8 \times 10^{-6}$) (Fig. 5C, Supplementary Data 15). Similarly, a lower expression ratio of *ULK3*-201 was inferred to be causally associated with a lower level of lymphocyte percentage, consistent with the GWAS finding[41] (MR beta = 0.054, $P = 1.4 \times 10^{-12}$). These findings support a model in which downregulation of *ULK3*-201 contributes to lower levels of DBP and lymphocyte percentage, highlighting a mechanistic connection between isoform-specific regulation and cardiovascular traits.

The *ULK3* region (±250 kb) harbors multiple *cis*-irQTLs and *cis*-eQTLs, with rs12898397 identified as the sentinel irQTL and rs936228 as the sentinel *cis*-eQTL[35] using the FHS dataset (Fig. 5D, Supplementary Data 16). Overall, these *cis*-eQTLs displayed lower statistical significance than the *cis*-irQTLs. The sentinel *cis*-eQTL (rs936228) is located approximately 2 kb from the sentinel *cis*-irQTL (rs12898397),

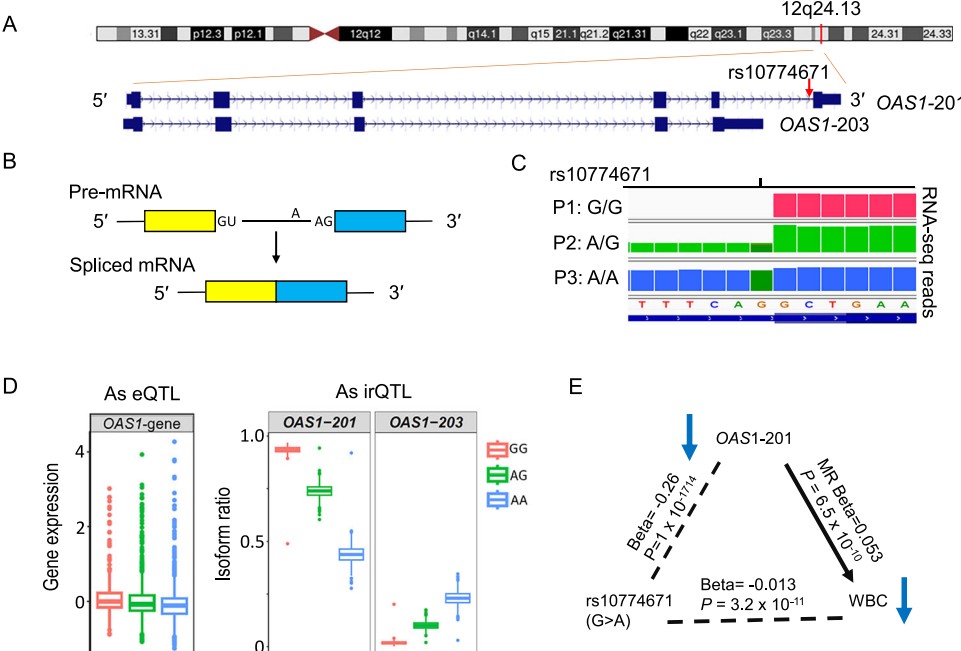

**Fig. 4 | Analyses of irQTLs Illustrated with the *OAS1* Gene. A** *OAS1* on the positive strand of chromosome 12 (hg38). Its pre-mRNA undergoes editing and alternative splicing. The sentinel irQTL, rs10774671 (G>A), lies at the splice acceptor site of intron 5. **B** Schematic of alternative splicing: introns with conserved donor (GU/GT) and acceptor (AG) sites are removed from pre-mRNA, generating isoforms with defined splice junctions. **C** IGV visualization of RNA-seq reads near rs10774671 (G>A) shows genotype-dependent splicing in three FHS participants: G/G with exon skipping, A/G with partial splicing, and A/A with full intron 5 splicing. These patterns align with high, intermediate, and low expression of isoform *OAS1*-201 (ENST00000202917.10). **D** Overall *OAS1* expression is weakly associated with rs10774671, showing a decrease with increasing A allele dosage ("as eQTL"). The box plots show median (IQR; range) expression as −0.0027 (−0.16, 0.23; −0.89, 3.0) for G/G (*n* = 386), −0.074 (−0.25, 0.17; −1.09, 3.93) for A/G (*n* = 1,172), and −0.11 (−0.33, 0.081; −1.26, 4.28) for A/A (*n* = 999). In contrast, isoform ratios of *OAS1*-201

and *OAS1*-203 are strongly associated with rs10774671 (G>A) as irQTLs, with *OAS1*-201 decreasing and *OAS1*-203 increasing with A allele dosage. The median (IQR; range) isoform ratio of *OAS1*-201 was 0.94 (0.93, 0.95; 0.49, 0.97) for G/G, 0.74 (0.72, 0.76; 0.60, 0.94) for A/G, and 0.44 (0.41, 0.46; 0.28, 0.92) for A/A. The median (IQR; range) isoform ratio of *OAS1*-203 was 0.018 (0.014, 0.023; 0.0015, 0.20) for G/G, 0.10 (0.089, 0.113; 0.020, 0.18) for A/G, and 0.231 (0.209, 0.252; 0.030, 0.35) for A/A. **E** The rs10774671-A allele is associated with reduced *OAS1*-201 expression in linear regression ($P = 1 \times 10^{-1714}$). MR supports a causal relationship between lower *OAS1*-201 levels and reduced WBC count ($P = 6.5 \times 10^{-10}$), consistent with GWAS findings associating rs10774671-A with lower WBC levels ($P = 3.2 \times 10^{-11}$). All tests are two-sided *t*-test or *z*-test (for MR) in regression analyses. IGV Integrative Genomics Viewer, IQR interquartile range, WBC white blood cell, MR Mendelian randomization. Gene names are italicized.

with moderate LD with each other (LD $R^2 = 0.62$). Of note, rs936228 was associated with comorbidity of COVID-19 critical illness and hypertension[45]. Additional SNPs nearby that are in strong ($R^2 > 0.9$) or moderate ($0.61 < R^2 < 0.63$) LD with the two sentinel QTLs exhibited significant associations with several traits in GWAS (Supplementary Data 16). Future studies are needed to investigate the role of irQTLs and eQTLs in trait associations.

To assess the impact of the rs12898397 A>G variant on *ULK3* splicing and structure, we applied MaxENT to the 9 nucleotides around the variant, the exon 14 (Fig. 6A)[46], which predicts splice site strength using entropy models[46]. By prediction, the rs12898397-G allele weakens the canonical 5′ GT donor site (MaxENT score: 7.93 → 4.44) while strengthens an alternative GT site within the exon (MaxENT score: 2.98 → 7.20), promoting skipping of six nucleotides (GTC-AAG) and loss of two amino acids ("VK") (Fig. 6B). To validate this allele-dependent splicing pattern, we analyzed RNA-seq reads from three randomly selected FHS participants with A/A, A/G, and G/G genotypes (Fig. 6C). As expected, the A/A individual used the canonical TG donor site to splice intron 14, while the G/G individual utilized a cryptic TG site, leading to loss of six nucleotides in the mature mRNA. The individual with A/G showed intermediate splicing, with ~50% of reads retaining the six bases. To assess structural impact, we used AlphaFold 3, an AI-powered tool with high prediction accuracy[47,48]. The Google AlphaFold 3 analysis suggests conformational changes in the 173-amino acid C-terminal region, likely reflecting the VK loss (Fig. 6D, Supplementary Figs. 6 & 7).

## Distinct irQTLs associated with multiple isoform transcripts of *CNN2*

*CNN2* (calponin 2, 19p13.3) encodes a regulatory protein that interacts through its three well-conserved 26 amino acid residue domains with actin, tropomyosin, and other cellular components, to modify cytoskeleton dynamics[49,50]. *CNN2* consists of five exons (Fig. 7A), producing ten detectable isoform transcripts in FHS (Supplementary Data 14). We found two significant sentinel *cis*-irQTLs (Supplementary Data 3), rs930232 (G>A) and rs5014188 (T>C) (pairwise LD, $R^2 = 0.13$, all population in LDlink[51]), that were linked to three *CNN2* isoforms, ENST00000263097.9 (*CNN2*-201), ENST00000348419.7 (*CNN2*-202), and ENST00000568865.3 (*CNN2*-209) (Fig. 7A).

The rs930232-A allele was strongly associated with a lower isoform ratio of *CNN2*-201 ($R^2 = 0.61$, beta = −0.032, $P = 1 \times 10^{-532}$) and with a higher ratio of *CNN2*-202 ($R^2 = 0.65$, beta = 0.030, $P = 1 \times 10^{-586}$); however, it was only weakly associated with *CNN2*-209 ($R^2 = 0.04$, beta = −0.00064, $P = 1 \times 10^{-27}$) (Fig. 7B, Supplementary Data 3). Despite being a strong irQTL, the rs930232 (G>A) variant was not predicted to directly influence splice sites by either SpliceAI or MaxENT. The rs930232-A allele was associated with a higher neutrophil count (beta = 0.021, $P = 1 \times 10^{-26}$) in 746,667 individuals in a blood-cell composition GWAS[52] (Fig. 7C). In FHS participants, the *CNN2*-201 isoform ratio was negatively associated with neutrophil count (beta = −0.00042, $P = 0.0015$); MR also found that the *CNN2*-201 isoform ratio was negatively associated with neutrophil count (beta = −0.50, $P = 2 \times 10^{-15}$) (Fig. 7C, Supplementary Data 15).

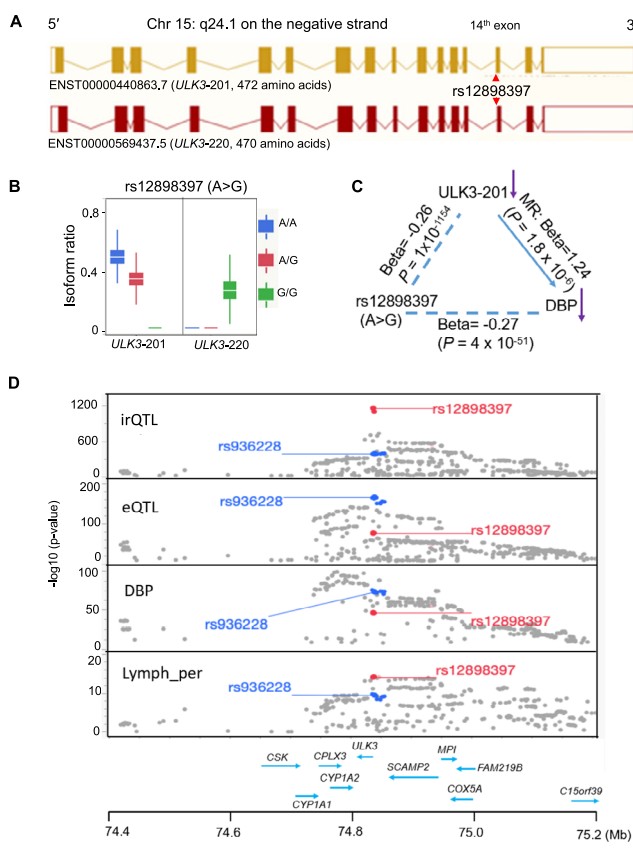

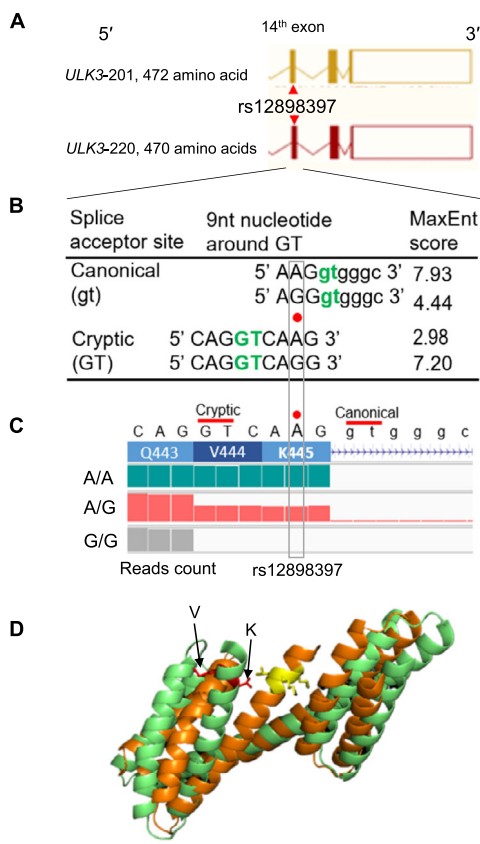

**Fig. 5 | Genetic associations of rs12898397-G with *ULK3* splicing, expression, and blood pressure trait. A** unc-51-like kinase 3 (*ULK3*), containing 16 exons, is located on the negative strand of chromosome 15. To maintain $5' \to 3'$ transcriptional orientation, the positive-strand sequence around rs12898397 (e.g., T > C) was converted to its complementary negative-strand form (A > G). **B** rs12898397 (A > G) showed the strongest association with the isoform ratio of *ULK3*-201 (ENST00000440863.7), which encodes the full-length 472-aa protein ($\beta = -0.26$, $P = 1 \times 10^{-1154}$). Median (IQR; range) isoform ratios were 0.49 (0.44, 0.54; 0.25, 0.84) for A/A ($n = 444$), 0.34 (0.30, 0.39; 0, 0.62) for A/G ($n = 1,208$), and 0 (0, 0; 0, 0.34) for G/G ($n = 999$). In contrast, rs12898397 was positively associated with *ULK3*-220 (ENST00000569437.5) (beta = 0.16, $P = 1 \times 10^{-573}$), which lacks two amino acids ('VK') at the 5' end of exon 14. Median (IQR; range) isoform ratios were 0 (0, 0; 0, 0.011) for A/A, 0 (0, 0; 0, 0.49) for A/G, and 0.26 (0.20, 0.33; 0, 0.69) for G/G. **C** MR supports a causal association ($P = 1.8 \times 10^{-6}$) between reduced *ULK3*-201 level and DBP ($P = 1 \times 10^{-1154}$), consistent with the GWAS association of rs12898397-G ($P = 4 \times 10^{-51}$). **D** This region harbors multiple *cis*-irQTLs and *cis*-eQTLs, each showing strong associations with DBP and lymphocyte percentage (lymph_per). The sentinel irQTL (rs12898397) and eQTL (rs936228, $P = 1.4 \times 10^{-165}$) within *CYP1A1*−*ULK3* region are associated with both DBP (rs12898397: $P = 9.2 \times 10^{-47}$; rs936228: $P = 1.1 \times 10^{-73}$) and lymphocyte percentage (rs12898397: $P = 8.1 \times 10^{-15}$; rs936228: $P = 3.6 \times 10^{-10}$). However, the underlying genetic mechanisms connecting these QTLs to the observed GWAS signals remain to be elucidated. All tests are two-sided *t*-test or *z*-test (for MR) in regression analyses. IQR interquartile range, DBP diastolic blood pressure, GWAS genome-wide association study, MR Mendelian randomization. Gene names are italicized.

**Fig. 6 | Allele-dependent splice donor usage associated with rs12898397 in *ULK3*. A** The *ULK3* exon 14 region is shown in transcriptional orientation ($5' \to 3'$) as the complementary positive-strand sequence, highlighting two isoform transcripts, *ULK3*−201 and *ULK3*-220. **B** MaxENT was applied to 9-nucleotide sequences surrounding rs12898397. Lowercase letters denote intronic sequences and capital letters denote exon sequences. The rs12898397-G allele was predicted to weaken the canonical splice donor site (gt: $7.93 \to 4.44$) and strengthen a cryptic donor site (GT: $2.98 \to 7.20$), supporting allele-dependent splice switching. The green color highlights the canonical and cryptic splice donor sites. **C** RNA-seq from three FHS subjects showed rs12898397 (A > G) genotype-dependent splicing: A/A (green) used the canonical donor, G/G (gray) used a cryptic donor skipping two codons (GTC-AAG), resulting in the absence of the valine-lysine ("VK") dipeptide, and A/G (pink) showed mixed splicing. Lowercase letters denote intronic sequence. **D** PyMOL visualization of AlphaFold 3.0-predicted C-terminal models of *ULK3* isoforms, with (173 amino acids in green color) and without (in orange color) the "VK" dipeptide, showed substantial structural divergence when aligned. The yellow region in the model without "VK" highlights the area without "VK". See Supplementary Figs. 6 & 7 for details. The Root-Mean-Square Deviation (RMSD), which quantifies the average atomic distance between superimposed protein structures, was 5.81 Å, indicating substantial structural divergence. In general, RMSD values above 2–3 Å suggest significant conformational differences between compared models. Gene names are italicized.

Another *cis*-irQTL of *CNN2*, rs5014188-C, was strongly associated with a lower relative level of *CNN2*-209 ($R^2 = 0.58$, beta = −0.0034, $P = 1 \times 10^{-481}$) while it showed no significant association with *CNN2*-201 ($P > 5 \times 10^{-8}$) and a much weaker, positive association with *CNN2*-202 ($R^2 = 0.02$, beta = 0.0081, $P = 1 \times 10^{-13}$) (Fig. 7B). The effect of *CNN2* isoform variation on predicted protein sequence is depicted in Fig. 7D. Predicted protein sequences were based on the standard translation of each of the mRNA sequences in the UCSC Genome Browser database[53] (https://genome.ucsc.edu/index.html). Three well-known calponin-like repeat motifs are indicated, superimposed on the contributions

from exons 5, 6, and 7 in the three protein isoforms[49]. Loss of exon 5 in *CNN2*-202 truncates the first repeat, while the gain of exon 7 in *CNN2*-209 truncates the second repeat and obliterates the third repeat entirely[49]. These motifs are essential to the actin-binding properties of calmodulin 2, and thus, modifications of the relative amounts of these three isoforms, *CNN2*-201, *CNN2*-202, and *CNN2*-209, may mediate observed phenotypic consequences of *CNN2* irQTLs.

## Discussion

We conducted comprehensive analyses using whole-genome and whole-transcriptome sequencing to identify genetic variants associated with isoform variation. Using whole-blood-derived DNA for WGS and RNA for RNA-seq in 2622 FHS participants, we identified

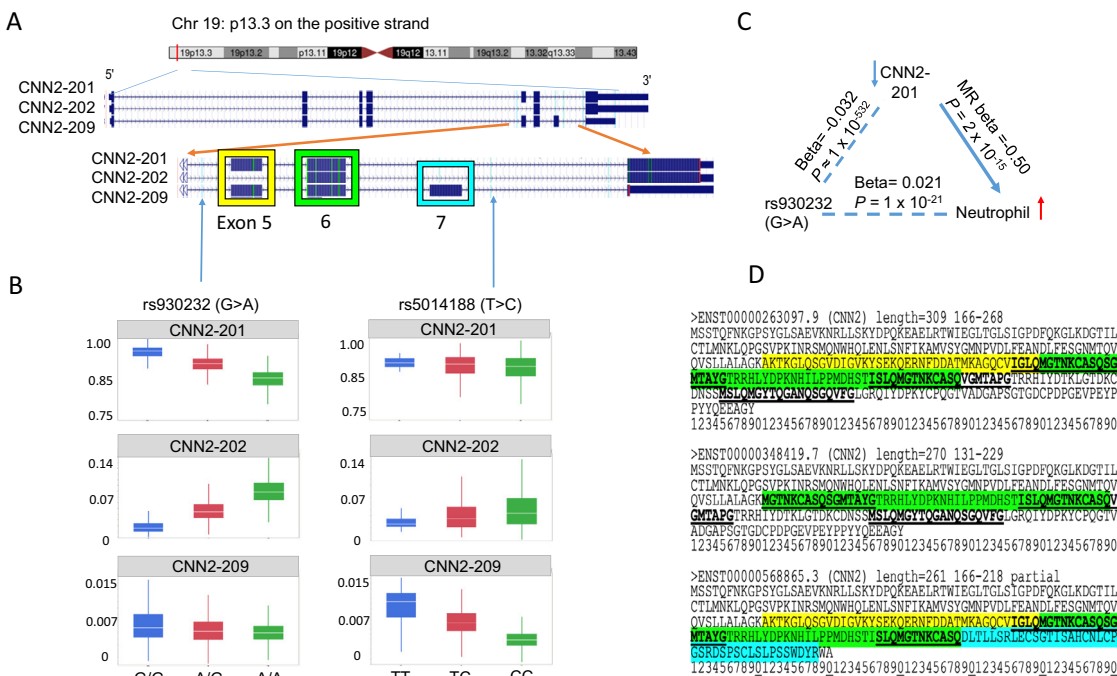

**Fig. 7 | CNN2 transcripts under genetic control. A** Genome browser view of *CNN2* on the positive strand of chromosome 19 (hg38). Three *CNN2* isoforms, *CNN2*-201 (ENST00000263097.9), *CNN2*-202 (ENST00000348419.7), and *CNN2*-209 (ENST00000568865.3), differ by the inclusion of Exons 5 and 6, Exon 6 alone, or Exons 5, 6, and 7, respectively. **B** Isoform ratios of *CNN2*-201 and *CNN2*-202 are strongly associated with the sentinel *cis*-irQTL rs930232, while *CNN2*-209 is more strongly associated with a different *cis*-irQTL, rs5014188, in regression analysis. For rs930232, median (IQR; range) isoform ratios were as follows: for *CNN2*-201, 0.91 (0.90–0.92; 0.51–0.94) in G/G (*n* = 781), 0.88 (0.86–0.89; 0.35–0.93) in A/G (*n* = 1309), and 0.84 (0.83–0.86; 0.40–0.90) in A/A (*n* = 532); for *CNN2*-202, 0.015 (0.010–0.024; 0–0.33), 0.044 (0.033–0.057; 0–0.42), and 0.079 (0.065–0.095; 0.016–0.50), respectively; and for *CNN2*-209, 0.0058 (0.004–0.008; 0–0.017), 0.0052 (0.004–0.007; 0–0.019), and 0.0050 (0.004–0.006; 0–0.012). For rs5014188, median (IQR; range) isoform ratios were: for *CNN2*-201, 0.89

(0.88–0.90; 0.51–0.91) in T/T (*n* = 52), 0.89 (0.86–0.90; 0.56–0.94) in C/T (*n* = 534), and 0.88 (0.85–0.90; 0.35–0.94) in C/C (*n* = 2036); for *CNN2*-202, 0.026 (0.022–0.034; 0.012–0.33), 0.035 (0.020–0.054; 0.004–0.33), and 0.043 (0.026–0.069; 0–0.50), respectively; and for *CNN2*-209, 0.012 (0.0093–0.014; 0.0024–0.017), 0.0082 (0.0066–0.010; 0–0.019), and 0.0048 (0.0036–0.0060; 0–0.011). **C** rs930232 (G>A) is associated with reduced expression of *CNN2*-201 (beta = −0.032, $P = 1 \times 10^{-532}$), which is causally linked to higher neutrophil percentage by Mendelian randomization (beta = −0.50, $P = 2 \times 10^{-15}$), consistent with GWAS results showing a positive association between rs930232 and neutrophil percentage (beta = 0.021, $P = 1 \times 10^{-21}$). (D) Predicted calponin 2 protein sequences show how Exons 5, 6, and 7 contribute to isoform structure. The three conserved 26-residue actin-binding domains are underlined in each isoform. All tests are two-sided *t*-test or *z*-test (for MR) in regression analyses. IQR interquartile range, MR Mendelian randomization. Gene names are italicized.

---

10,883 isoforms across genes associated with at least one significant *cis*-irQTL ($P < 5 \times 10^{-8}$) with MAF ≥ 0.01, and 1084 isoforms across 590 genes associated with at least one significant *trans*-irQTL ($P = 1.5 \times 10^{-13}$) with MAF ≥ 0.01. These associations replicated well, with *cis*-irQTL replication rates of 76.7% in FHS and 72.4% in WHI, and *trans*-irQTL replication rates of 80.8% and 64.3%, respectively. Together, these results provide a large-scale catalog of regulatory variants influencing isoform usage and establish a resource for understanding the genetic architecture of splicing regulation.

The isoform ratio method[18] captures the relative expression of each isoform within a gene, allowing the detection of regulatory variation that affects full-length splice isoforms. It is particularly useful for identifying biologically interpretable changes, such as isoform switches that impact coding potential, untranslated regions, or transcript stability. Compared to exon-level QTLs[21], which detect local events like exon skipping, and junction-based LeafCutter QTLs[16,17], which are optimized for discovering novel or cryptic intron excision, the isoform ratio method provides a clearer view of transcript-level consequences (Supplementary Data 17). However, it relies on comprehensive and accurate transcript annotations, potentially missing events in poorly annotated genes. It also has a lower resolution to identify specific splice sites or exons, which may overlook subtle or complex splicing changes. Additionally, the isoform ratio method introduces interdependency among isoforms of the same gene, as their proportions must sum to 1. Consequently, an increase in the expression ratio of one isoform may correspond to a decrease in one

or more others. Despite these limitations, its high biological interpretability and relevance to phenotype-driven studies make it a valuable complement to other QTL approaches.

To address these limitations and better characterize isoform ratio QTLs (irQTLs), we integrated multilevel computational annotations including transcript structure, splicing predictions, variant effects, and regulatory data from established genomic databases[29–32]. This approach enabled functional interpretation of irQTLs acting at both canonical and non-canonical splice sites. For example, the *cis*-irQTL rs10774671 (G>A) in *OAS1* disrupts a 3′ splice acceptor site (predicted by SpliceAI[30]), changing expression from the antiviral isoform *OAS1*-p46 to the less effective *OAS1*-p42, potentially modulating COVID-19 severity by altering how *OAS1* splicing influences white blood cell counts[39,54]. MR analysis, along with GWAS[52], supports a causal role for this splicing event in immune regulation[38–40]. Similarly, rs12898397 (T>C) in *ULK3* alters splice site usage (predicted by MaxEnt[46]), resulting in exon skipping and a truncated protein, and a causal link to DBP, with support from GWAS[55] and MR analyses. Finally, irQTLs at the *CNN2* locus showed complex exon inclusion patterns not predicted by existing tools, suggesting regulation via non-canonical mechanisms.

In addition to FHS, we analyzed community-based cohorts, JHS[22] and WHI[23], to enable independent replication and enhance the generalizability of our findings across diverse populations. We observed a 72% replication rate for *cis*-irQTLs in WHI, supporting the robustness of our results. In contrast, replication in JHS was lower (32%), likely reflecting both technical and biological differences. Notably, JHS used

RNA-seq from PBMCs, while FHS and WHI used whole blood. To further assess the impact of tissue source and ancestry, we stratified WHI by ancestry. Replication was highest in the full WHI cohort and lower in ancestry-specific subsets, partly reflecting reduced sample size (Supplementary Fig. 2). Despite similar sample sizes, replication in JHS African American participants was lower than in WHI African American participants, suggesting that tissue source may have a greater effect on replication than ancestry. While clinical characteristics also varied across cohorts (Table 1, Supplementary Data 2), their influence on splicing could not be directly assessed. These results underscore the importance of considering tissue type, ancestry, and sample size in cross-cohort transcriptomic analyses.

Using our approach, cis-irQTLs showed higher external replication rates than trans-irQTLs in WHI ($n = 2005$), consistent with published findings[56,57], likely reflecting their proximity to target isoforms and involvement in local regulation. Nevertheless, we observed more replicated trans-irQTL-isoform pairs than cis pairs in the additional FHS sample ($n = 1094$), which may be explained by consistent experimental conditions and shared analytical pipelines across FHS discovery and replication sample. Our study confirmed a subset of splicing QTL-gene pairs previously reported in the FHS[21], identifying concordant associations for 17.3% of the 2650 pairs tested ($P < 1 \times 10^{-4}$). The limited overlap may reflect differences in study design and methodology, including genetic and transcriptomic platforms (imputed SNPs vs. WGS; exon arrays vs. RNA-seq) and the expression features tested (exon-level expression vs. isoform ratios). Some previously reported associations may also represent false positives, highlighting the importance of independent validation with high-resolution data. Our study has limited power to detect rare irQTLs reflecting constraints related to sample size and allele frequency thresholds. Larger and more diverse cohorts, along with alternative methods (e.g., SKAT[58] and SKAT-O[59]), will be needed to capture the full spectrum of rare splicing regulatory variation.

In summary, our study presents a comprehensive map of genetic regulation at the isoform level, offering additional insights into the complexity of transcriptomic variation. By integrating isoform ratio modeling, splicing prediction tools, and causal inference analyses, we demonstrate that irQTLs play a distinct and functionally significant role in potentially mediating genetic associations with human traits. These findings emphasize the role of transcript-level analysis in understanding the mechanisms through which genetic variation influences phenotype. Future research should prioritize systematic functional characterization[60,61] of irQTLs and clarify their unique contributions to variant-to-trait pathways relative to eQTLs.

## Methods

### Ethical compliance
All research was conducted in accordance with relevant ethical guidelines and regulations. Study protocols were approved by the Institutional Review Boards (IRBs) at participating sites: the Boston Medical Center and Boston University Medical Campus IRB (FHS); Jackson State University, Tougaloo College, and the University of Mississippi Medical Center IRBs (JHS); and the IRBs of all participating institutions, with oversight from the Fred Hutchinson Cancer Research Center IRB (WHI). All participants provided written informed consent for genetic studies.

### Study participants
We included 3716 participants ($n = 2622$ for initial discovery and $n = 1094$ for internal replication), mostly self-identified White American participants primarily from the FHS Offspring and Third Generation cohorts (Table 1)[62,63]. Blood samples for RNA-seq were collected at the ninth examination cycle (2011–2014) of the Offspring cohort and the second examination cycle (2008–2011) of the Third Generation cohort (Table 1). In WHI, blood samples for RNA-seq were collected

during a follow-up examination (2012–2013) conducted as part of the Long Life Study (LLS), a sub-study within the larger WHI cohort[23]. The LLS repeated selected components of the original WHI physical examination and collected blood samples from approximately 8000 participants across the United States. A total of 2005 WHI participants were included in the present analysis, comprising 1000 White American participants, 918 African American participants, and 87 participants who self-identified as other race/ethnicity groups. We did not exclude any participants based on genetic similarity to reference panels such as 1000G. We also analyze 1010 self-identified African American participants from JHS (Supplementary Information).

This study included 54.5% women in FHS, 100% in WHI, and 64.4% in JHS, based on self-reported sex (Table and Supplementary Data 2). Sex distribution was not part of the study design, as the cohorts were previously established as population-based studies. No sex-specific analyses were performed.

### Isolation of RNA from whole blood
Isolation of RNA from peripheral whole blood[35,56] was described briefly below. PAXgene™ tubes (PreAnalytiX, Hombrechtikon, Switzerland) were used to collect peripheral whole-blood samples (2.5 mL) from both FHS and WHI participants. Total RNA was extracted using a standard protocol with the PAXgene Blood RNA Kit, either at local study-specific laboratories or at the TOPMed contract laboratory at the Northwest Genomics Center. After centrifugation and washing, white blood cell pellets were collected and lysed in a guanidinium-containing buffer. The extracted RNA was evaluated for quality by determining absorbance readings at 260 and 280 nm using a NanoDrop ND-1000 UV spectrophotometer. The integrity of total RNA was determined with the Agilent Bioanalyzer 2100 microfluidic electrophoresis (Nano Assay and the Caliper LabChip system). In JHS, RNA was extracted from peripheral blood mononuclear cells (PBMC) samples. Detailed blood sample collection and RNA extraction for JHS were described in the Supplementary Information. Given the difference in RNA sources (i.e., whole blood vs. PBMC), we used JHS as the secondary replication cohort.

### Whole-genome sequencing
Details on DNA extraction, processing, and identification of variants[22] were described briefly below. Whole-genome sequencing (WGS) was conducted in one of the TOPMed contract sequencing centers. Consistent sequencing and data processing criteria were used across all centers, with DNA sequence alignment to the human genome build GRCh38. The resulting BAM files were sent to TOPMed's Informatics Research Center (IRC) for realignment and standardized BAM file generation. This study used WGS data from Freeze 10a, which included approximately 208 million SNPs across TOPMed cohorts.

### RNA sequencing and processing
RNA sequencing was conducted at one of the NHLBI TOPMed program contract laboratories, following a standard protocol[35] for data processing. Base call generation was performed on the NovaSeq 6000 instrument (RTA 3.1.5), followed by the conversion of demultiplexed, unaligned BAM files to FASTQ format using SamTools bam2fq (v1.4). Sequence and base quality were assessed with the FASTX-toolkit (v0.0.13), and sequence alignment to GRCh38 with reference transcriptome GENCODE release 30 was conducted using STAR. FastQC (version 0.11.9) provided initial quality control metrics. A genome index for mapping was built by STAR to generate BAM files at the gene and transcript levels based on the GRCh38 reference build. The TOPMed RNA-seq pipeline utilized RNA-SeQC[64] to derive standard quality control metrics from aligned reads (Supplementary Information, Supplementary Fig. 8). Consistent normalization was conducted across the FHS, JHS, and WHI samples.

## Calculation of isoform ratio

Transcript-level expression of an isoform was quantified as Fragments Per Kilobase of transcript per Million mapped reads (FPKM), a normalization measure that considers the gene length in quantifying the expression level of a transcript. The trimmed means of M-values (TMM)[65] were derived from the FPKM values using the edgeR package[64,66]. The TMM method estimates scale factors between samples and relative RNA production levels from RNA-seq data. The isoform expression ratio was calculated as the transcript level of an isoform (TMM value) divided by the sum of the levels of all isoforms (the sum of the TMM values) of its parent gene. The transcript isoform ratio was considered as missing if the overall expression level of its gene was zero[18].

## Association analysis of genetic variants with isoform ratios

We performed linear regression analyses of genome-wide SNPs having minor allele frequencies (MAFs) $\geq 0.01$ and Hardy−Weinberg Equilibrium (HWE) $p \geq 1e\text{-}10$ with isoform ratios for transcripts having the population median TMM $\geq 2$ and fewer than 20% missing values. The dosage (0, 1, or 2) of the alternative allele for each SNP was used as a continuous independent variable (additive model), with the isoform ratio as the dependent variable, adjusting for age, self-reported sex (not in WHI), white blood cell counts, and additional variables described below. In FHS, the isoform ratio was adjusted for family relationships to obtain residuals, which were then used as the dependent variable, while in WHI, the isoform ratio was used directly in the regression model. To minimize confounding, regression models included five genotype-based principal components (PCs) to account for population stratification and 15 transcriptome-based PCs to adjust for unknown confounders affecting gene expression. We explored additional technical covariates, including year of blood collection, batch (sequencing machine and time, plate and well), and RNA concentration[35] but these covariates were not included in the final models since they explained <1% of variance in isoform ratios. We employed the NIH-supported STRIDES cloud computing infrastructure for all association analyses. We used a Graphical Processing Unit (GPU)-based program[56] for all computations. We stored effect sizes, standard error, partial R-squared, and $p$-values for SNP-isoform pairs with $P < 1 \times 10^{-4}$ to facilitate future meta-analysis and lookups by the broad scientific community.

We identified significant isoform ratio QTLs (irQTL). A $cis$-irQTL was defined as a polymorphism falling within 1 Mb of the transcription start site (TSS) of the isoform. In contrast, a $trans$-irQTL was defined as a variant located more than 1 Mb from the TSS or on a different chromosome. We used $P < 5 \times 10^{-8}$ to denote a significant $cis$-irQTL and $P < 1.5 \times 10^{-13}$ (-0.05/(25,642 transcripts × 12,887,093 SNPs)) to define a significant $trans$-irQTL.

## Internal and external replication

Association analyses were conducted in two stages: an initial discovery stage ($n = 2622$), followed by a replication (Table 1). We conducted replication on the irQTL-isoform pairs from sentinel irQTLs (the most significant irQTL per isoform). Internal replication was performed in an additional FHS sample ($n = 1094$), while external replication was performed in the WHI cohort ($n = 2005$, primary external replication) and the JHS cohort ($n = 1010$, secondary external replication, given the use of a different RNA-seq source) (Table 1, Supplementary Information, Supplementary Data 2 & 7). In WHI, we conducted analyses in the full sample, White American participants ($n = 1000$), and African American participants ($n = 918$). Replication was assessed only for irQTL-isoform pairs that were present in both FHS and the respective replication cohort. An irQTL-isoform pair was considered successfully replicated if it had a $P < 1e\text{-}4$ (for both $cis$- and $trans$- pairs) and consistent effect direction (i.e., the same sign of beta estimates) between the discovery and replication data. The replication rate was calculated as the

proportion of irQTL-isoform pairs meeting these criteria among all pairs that were present in both the discovery and the replication samples. We calculated the replication rate for irQTL-isoform pairs from sentinel $cis$-irQTLs and $trans$-irQTLs. A sentinel $cis$-irQTL was defined as the variant most strongly associated with an isoform (i.e., lowest $p$-value) within 1 Mb. For each isoform, a sentinel $trans$-irQTL was identified on each of the 23 chromosomes (22 autosomes and the X chromosome).

## Comparison of functional relevance of $cis$-irQTLs and $cis$-eQTLs

We conducted several analyses to explore the functional relevance of $cis$-irQTLs in comparison to $cis$-eQTLs[35] in the same participants. We compared the number of genes containing $cis$-irQTLs to those containing $cis$-eQTLs. We also compared the positions of sentinel $cis$-eQTLs and $cis$-irQTLs relative to the 5' start and 3' end of their associated genes. Additionally, we evaluated the variance explained by $cis$-irQTLs compared to that explained by $cis$-eQTLs.

We assessed the enrichment of sentinel $cis$-irQTLs and $cis$-eQTLs in relevant genomic features, including splice donor/acceptor sites, intron/exon boundaries, and eCLIP peaks for binding sites of 203 RNA-binding proteins (RBPs) from ENCODE[67,68]. To reduce bias, we generated 1000 sets, consisting of randomly selected SNPs with MAF $\geq 1\%$ and Hardy−Weinberg equilibrium $P \geq 1 \times 10^{-10}$, one for each $cis$-irQTL or $cis$-eQTL, matched based on similar MAF and distance to the TSS for the respective $cis$-irQTLs or $cis$-eQTLs, using the following bin boundaries: MAF bins at 0.01, 0.05, 0.1, 0.2, 0.3, and 0.5, and TSS distance bins at 0 kb, 1 kb, 10 kb, 100 kb, and 1 Mb. Enrichment of $cis$-irQTLs and $cis$-eQTLs in each feature was calculated as follows:

$$sQTL\ enrichment = \frac{proportion\ of\ observed\ top\ sQTL\ with\ a\ genomic\ feature}{Average\ proportion\ of\ random\ SNPs\ with\ a\ genomic\ feature}$$

$$eQTL\ enrichment = \frac{propotion\ of\ observed\ top\ eQTL\ with\ a\ genomic\ feature}{Average\ proportion\ of\ random\ SNPs\ with\ a\ genomic\ feature}.$$

## Enrichment analysis of $cis$-irQTLs with GWAS

**Overlap of irQTLs with GWAS Catalog SNPs**. We selected GWAS Catalog SNPs[25] using two criteria: (1) a significance threshold of $P < 5 \times 10^{-8}$, and (2) a sample size threshold reported with both discovery and replication sample sizes $\geq 5000$, or the discovery sample size $\geq 10,000$. Replicated irQTLs were those in the irQTL-isoform pairs with $P < 5 \times 10^{-8}$ in the discovery sample. We then matched the selected GWAS Catalog SNPs with the identified irQTLs.

**Enrichment analyses using Fisher's exact test**. We tested the null hypothesis that irQTLs are independent of GWAS SNPs (i.e., not enriched among them). A total of 12,887,093 SNPs across the genome (MAF $\geq 1\%$ and Hardy−Weinberg equilibrium $P \geq 1 \times 10^{-10}$) served as the reference set. For example, we constructed a 2 × 2 contingency table including: (1) the number of SNPs overlapping between GWAS SNPs and irQTLs, (2) SNPs present in GWAS but not in irQTLs, (3) SNPs in irQTLs but not in GWAS, and (4) all remaining SNPs. Fisher's exact test was used in this analysis. We presented results with $P < 1 \times 10^{-4}$.

## Prediction of SNP effects on alternative splicing

We used two methods, SpliceAI[30] and MaxENT[46], to predict the SNP effects for the 25 top irQTLs on transcriptomic splice variation. SpliceAI uses a deep neural network to identify splicing and helps predict non-coding cryptic splice variants in the human genome[30]. MaxENT refers to maximum entropy models that are specified by changing a set of constraints to discriminate human DNA 5'GT (i.e., donor) and 3'AG (i.e., acceptor) splice sites from non-functional decoys[46].

### Prediction of protein structures by Google AlphaFold 3.0

We used Google AlphaFold 3.0 to predict protein structures[47,48] for the splicing variant rs12898397 [T>C] in *ULK3* [unc-51-like kinase 3] as an example of how such variants might affect protein structure (Supplementary Information). rs12898397 [T>C] is located in the COOH-terminal region of ULK3 (NM_001411082.1). To confirm the structure, we first aligned it with the reference 4WZX crystal structure from the Protein Data Bank[69] using the PyMOL Molecular Graphics System (Version 3.0.4, Schrödinger, LLC)[48]. Next, we predicted the structures of the COOH-terminal region in response to rs12898397 [T>C] using Google AlphaFold 3.0[47,48]. The overlay of these structures was visualized using PyMOL[48]. The structural differences were evaluated using root-mean-square deviation (RMSD) values, quantified in Angstrom units (Å). An Å of 1.0 or lower indicates nearly identical conformations, while an Å of 3.0 or higher suggests significant structural differences[70].

### Mendelian randomization (MR) analysis

To illustrate the utility of our irQTL resource, we conducted MR analysis to infer causality between the isoform ratios (as the exposure) of the respective transcripts for selected isoforms that harbor significant *cis*-irQTLs and CVD factors and blood composition traits (as the outcome). *cis*-irQTLs served as instrumental variables (IVs). We used the MR-base[26] software to conduct the MR analysis in conjunction with summary statistics of SNPs from a large GWAS database[25] (Supplementary Data 18). To minimize bias, we selected SNPs in MR analysis based on their linkage disequilibrium (LD) $r^2 < 0.001$[26]. We applied the inverse variance weighted (IVW) method to summarize results in MR analyses and used the MR-Egger method to assess SNP horizontal pleiotropic effects[26].

All statistical analyses were conducted using R (version 4.4.0) or JMP™ (version 18.1.1). Throughout the manuscript, we reported small *p*-values as −logP, representing $-\log_{10}$ (*p*-value).

### Reporting summary

Further information on research design is available in the Nature Portfolio Reporting Summary linked to this article.

## Data availability

The whole-genome sequencing (WGS), RNA sequencing (RNA-seq), and phenotypic data from the Framingham Heart Study (FHS), the Jackson Heart Study (JHS), and the Women's Health Study (WHS) used in this study are available at the dbGaP database under accession codes phs000007.v32.p13 (FHS) [https://www.ncbi.nlm.nih.gov/projects/gap/cgi-bin/study.cgi?study_id=phs000007.v34.p15], phs000964 (JHS) [https://www.ncbi.nlm.nih.gov/projects/gap/cgi-bin/study.cgi?study_id=phs000964.v5.p1], phs001237 (WHI) [https://www.ncbi.nlm.nih.gov/projects/gap/cgi-bin/study.cgi?study_id=phs001237.v3.p1]. The WGS, RNA-seq, and phenotypic data from all cohorts used in this study are available under restricted access to ensure confidentiality and protect participant privacy. Access can be obtained by submitting an ancillary study proposal and obtaining IRB approval. Timelines for the approval process range from 3–9 weeks. Results reported in this study are provided in the Supplementary information/Data accompanying this paper. Data for generating Figures (Figs. 2A, B, 3A, B, C, D, 4D, 5B, D, 7B; Supplementary Figs. 1–4) have been deposited at https://doi.org/10.6084/m9.figshare.29885138. All summary data generated in this study for irQTLs with $P < 1 \times 10^{-4}$ have been deposited at: https://www.ncbi.nlm.nih.gov/projects/gap/cgi-bin/study.cgi?study_id=phs001974.v8.p1. The data are accessible via dbGaP, with a straightforward application process to ensure secure and ethical use. Researchers must first obtain an eRA Commons account, which is typically set up within 3–7 business days if the applicant's institution is already registered with NIH (institutional registration may add an additional 2–4 weeks). Once the account is active, access to the data can be requested by submitting a Data Access Request (DAR) through dbGaP. DARs are reviewed on a biweekly basis, and approvals are generally granted within 2–4 weeks, provided the documentation is complete. Following approval, researchers can seamlessly log in to the NIH BioData Catalyst (BDC) platform using the same eRA Commons credentials to download or directly analyze the data, enabling timely and secure access for downstream research.

## Code availability

Codes for generating Figs. 2A, B, 3A, 3B, C, D, 4D, 5B, D, 7B; Supplementary Figs. 1–4 are available at https://doi.org/10.6084/m9.figshare.29885138.

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

## Acknowledgements

The Framingham Heart Study (FHS) was supported by NIH contracts N01-HC-25195, HHSN268201500001I, and 75N92019D00031. Molecular data for the Trans-Omics in Precision Medicine (TOPMed) program were supported by the National Heart, Lung, and Blood Institute (NHLBI). Whole-genome sequencing for NHLBI TOPMed: Whole-Genome Sequencing and Related Phenotypes in the Framingham Heart Study (phs000974.v1.p1) was performed at the Broad Institute Genomics Platform under grants 3R01HL092577-06S1 and 3U54HG003067-12S2. RNA-seq for the same project was performed at the Northwest Genomics Center under contract HHSN268201600032I. RNA-seq assays were supported in part by the Division of Intramural Research (PI D.L.) and an NIH Director's Challenge Award (PI D.L.). Centralized genomic read mapping, genotype calling, and variant quality control were provided by the TOPMed Informatics Research Center 3R01HL117626-02S1; HHSN268201800002I. The analytical component of this project was funded in part by the NHLBI Division of Intramural Research (D.L., R.J., T.H., S-J.H, C.Y.D), The Jackson Heart Study (JHS) is supported and conducted in collaboration with Jackson State University HHSN268201800013I, Tougaloo College HHSN268201800014I, the Mississippi State Department of Health HHSN268201800015I/HHSN26800001, and the University of Mississippi Medical Center HHSN268201800010I, HHSN268201800011I, and HHSN268201800012I, under contracts from the National Heart, Lung, and Blood Institute (NHLBI) and the National Institute on Minority Health and Health Disparities (NIMHD). We thank the staff and participants of the JHS. Genome sequencing for the JHS (phs000964.v1.p1) was performed at the Northwest Genomics Center HHSN268201100037C as part of the NHLBI Trans-Omics for Precision Medicine (TOPMed) program. Centralized genomic read mapping, genotype calling, and variant quality control were provided by the TOPMed Informatics Research Center 3R01HL117626-02S1; HHSN268201800002I. Phenotype harmonization, data management, sample identity quality control (QC), and program coordination were supported by the TOPMed Data Coordinating Center (R01HL120393; U01HL120393; HHSN268201800001I), and further supported through R01HL129132 (Y.L.), R01AG075884 (L.M.R.), and R01HL146500 (A.P.R.). The Women's Health Initiative (WHI) program is funded by the National Heart, Lung, and Blood Institute, National Institutes of Health, U.S. Department of Health and Human Services through contracts 75N92021D00001, 75N92021D00002, 75N92021D00003, 75N92021D00004, 75N92021D00005. The TOPMed WHI (phs001237) was supported by HHSN268201500014C and HHSN268201600034I (Broad Institute, WGS, RNA-seq, and metabolomics), HHSN268201600038I (Keck MGC, Methylomics). The RNA-seq data from WHI was also supported through TOPMed X01 HL139376 and X01HL153408. Scientific Computing Infrastructure at Fred Hutch was funded by ORIP grant S10OD02868, and partly supported by R01HL152439 (J.W.H., C.K., A.P.R.), R01HL175681 (N.F.), R01HG013163 (N.F.), R01HL163972 (N.F.), K26DK138425 (N.F.). No other authors received project-specific funding related to this study.

## Author contributions

C.L., R.J., J.M., P.M., and D.L. conceived the project. R.J., C.L., J.M., and P.M. performed statistical analyses. J.Y., M.W., and J.X. contributed to functional annotation analyses. N.L.H. and J.W.H. assisted with data acquisition. L.M.R., A.R., and Y.L. provided funding for sample processing, RNA sequencing, and analysis for JHS; G.O., J.M.M., and D.L. for FHS; and C.K. for WHI. C.L. wrote the manuscript. R.J., J.M., P.M., and D.L. reviewed and edited the manuscript. L.M.R., T.H., S.-J.H., J.W., Q.S., D.Y.C., P.O., N.F., P.L.A., and A.P.C. provided critical feedback. P.M. and D.L. supervised the project.

## Competing interests

L.M.R. is a consultant for the TOPMed Administrative Coordinating Center (through Westat). The remaining authors declare no competing interests.

## Additional information

[1]Department of Biostatistics, School of Public Health, Boston University, Boston, MA, USA. [2]Framingham Heart Study, Framingham, MA, USA. [3]Population Sciences Branch, Division of Intramural Research, National Heart, Lung, and Blood Institute, National Institutes of Health, Bethesda, MD, USA.

[4]Nutrition Epidemiology and Data Science, Friedman School of Nutrition Science and Policy, Tufts University, Boston, MA, USA. [5]Physiology and Pathophysiology, University of Manitoba, Winnipeg, MB, Canada. [6]Department of Genetics, University of North Carolina at Chapel Hill, Chapel Hill, NC, USA. [7]Department of Biostatistics, University of North Carolina at Chapel Hill, Chapel Hill, NC, USA. [8]Center for Computational and Genomic Medicine, Children's Hospital of Philadelphia, Philadelphia, PA, USA. [9]Department of Pathology and Laboratory Medicine, University of Pennsylvania, Philadelphia, PA, USA. [10]Critical Care Medicine Department, Clinical Center, National Institutes of Health, Bethesda, MD, USA. [11]Departments of Neurology, Boston University Schools of Medicine, Boston, MA, USA. [12]Department of Computational Medicine and Bioinformatics, University of Michigan, Ann Arbor, MI, USA. [13]Department of Medicine, University of Mississippi Medical Center, Jackson, MS, USA. [14]Division of Public Health Sciences, Fred Hutchinson Cancer Center, Seattle, WA, USA. [15]Department of Epidemiology, University of Washington, Seattle, WA, USA. [16]Department of Epidemiology, University of North Carolina at Chapel Hill, Chapel Hill, NC, USA. [17]Division of Biostatistics, Data Science Institute and Cancer Center, Medical College of Wisconsin, Milwaukee, WI, USA. [18]Pulmonary Center, Boston University School of Medicine, Boston, MA, USA. [19]Section of General Internal Medicine, Department of Medicine, Boston University Chobanian and Avedisin School of Medicine, Boston Medical Center, Boston, MA, USA. [20]These authors contributed equally: Chunyu Liu, Roby Joehanes, Jiantao Ma. ✉e-mail: liuc@bu.edu; roby.joehanes@nih.gov; levyd@nih.gov

