## [Peer Review file · Nature Communications]

Integrating Whole Genome and Transcriptome Sequencing to Characterize the Genetic Architecture of Isoform Variation

Corresponding Author: Dr Chunyu Liu

Version 0:

Reviewer comments:

Reviewer #1

(Remarks to the Author)

This study conducted a large-scale sQTL analysis on PBMC samples, identifying a substantial number of sQTL variants associated with isoform ratios. The manuscript is well-written, and the methodologies are generally applied appropriately. However, I believe there are several important concerns that need to be addressed by the authors to clearly distinguish the novelty of the findings.

1) I was surprised that the authors did not reference their previous sQTL study on the same FHS cohort, which utilized PBMC samples (Nat Genet 2015; doi: 10.1038/ng.3220). The authors should explicitly highlight the novel findings of the current study in comparison to their previous work. If this study represents a secondary analysis of the data, please clearly state this in the Methods section.

2) As suggested by the title, one of the strengths of this study appears to be the extensive whole-genome sequencing data. However, the authors did not fully capitalize on its potential benefits, as the analysis was limited to variants with a MAF > 0.01. The sQTL effects of rare variants could be assessed using alternative methodologies, such as evaluating allelic imbalance of isoform expression. If the number of variants is too large, in silico filtering methods like SpliceAI or MaxENT could be employed.

3) As discussed in the main text, the replication rate of sQTL signals was notably low at 23%. The authors speculated that this could be attributed to factors such as sample size, cellular population differences in PBMCs, and the distinct genetic ancestry of the replication samples. However, since the isoform ratio approach relies on the estimation of isoform expression using assembly tools, which may introduce biases, the possibility of false-positive findings cannot be ruled out. Therefore, I suggest that the authors validate the sQTL signals using junction-read-based approaches, such as LeafCutter.

4) The sQTL effects on isoform ratios can also be validated using long-read sequencing data, such as those provided by the GTEx project (Nature 2022; doi: 10.1038/s41586-022-05035-y).

5) Please discuss the potential reasons behind the observed enrichment of sQTL signals with pQTLs. Could it be that pQTL signals are influenced by the isoform ratio of genes? If the authors have any hypotheses regarding this, please provide specific examples or evidence to support them.

Reviewer #2

(Remarks to the Author)

In this study, the authors performed isoform ratio QTL mapping on a whole blood dataset, validated the associations on an external dataset, and analyzed the results with respect to functional characteristics, other molecular QTLs, and GWAS variants.

There are major issues with the methodology and presentation of this study that call into question the novelty and quality of the analyses presented.

1. There is a conflation of important concepts as demonstrated in the first sentence of the introduction: "Transcriptomic

isoform variation generates different messenger RNAs (mRNAs) from the same pre-mRNA, ...". This is only true of alternative splicing; other forms of isoform variation, namely alternative TSS and alternative polyA sites, are not generated from the same pre-mRNA.

Indeed, the presentation of the study and downstream analyses focus on alternative splicing, and the xQTLs are referred to as sQTLs, which they define as "splice variation quantitative trait locus". But the molecular phenotypes used are isoform ratios, which represent multiple types of variation including alternative splicing, alternative TSS, and alternative polyA sites, and there does not appear to have been any steps to narrow or process the isoforms so that they reflect alternative splicing specifically. It is therefore unclear to what degree the results of the splice-related analyses are supported by the "sQTL" data.

2. There is an existing study that performed "sQTL" mapping on the Framingham Heart Study dataset:

Zhang, X., Joehanes, R., Chen, B. et al. Identification of common genetic variants controlling transcript isoform variation in human whole blood. *Nat Genet* 47, 345–352 (2015). <https://doi.org/10.1038/ng.3220>

It is strange that this study was not cited or discussed in the present study, and that makes it difficult to assess the novelty of the present study. From a cursory examination, they appear similar.

3. There is heavy emphasis on the number of variant-phenotype pairs with significant association. The number of such associations is nearly meaningless, since it is well understood that in xQTL studies such as this, the vast majority of variants in the associated pairs are not involved in regulation, but rather are in LD with the causal variants. Similarly, the reported percentages of such pairs that were validated by the replication analysis is not meaningful, because even in different cohorts with different ancestries, variants in close proximity are likely to have some level of LD in both cohorts. Therefore, finding that high percentages of variant-phenotype pairs replicate does not indicate how many of the pairs actually have a real causal or functional relationship.

Other comments:

Line 128: "They found that expression levels of nearly all expressed genes (eGenes) are subject to genetic regulation." The cited GTEx study defines 'eGenes' as "genes with an eQTL", not "expressed genes".

The abstract mentions "allelic variation" twice in reference to a specific analysis, but doesn't define it, and the term is not used again in the text, so I can't tell if the analysis is actually presented. Perhaps it is just referring to the genotypic variation on which the QTLs are determined, but then this sentence:

"We identified over 3.5 million cis-sQTL-isoform pairs ($p < 5e-8$), comprising 1,176,624 cis-sQTL variants and 10,883 isoform transcripts from 4,971 sGenes, with significant change in isoform-to-gene ratio due to allelic variation."

would certainly be false, as the end of the sentence asserts a causal relationship, which is not the case for the vast majority of those 3.5 million cis-sQTL-isoform pairs.

Reviewer #3

(Remarks to the Author)

This manuscript identifies splice QTL in blood to better understand the genetic influences on alternative splicing and alternative splicing as a potential molecular mechanism for how GWAS SNPs influence disease. The authors use isoform ratios as their quantitative measure of alternative splicing and include both a within sample validation (i.e., split the original cohort into a discovery and validation set of samples) and a cross-population validation (i.e., samples from a completely independent cohort with many differences in sample demographics and in RNA isolation and processing). This is a well-thought-out study with a tremendous amount of data and impactful results. In general, major issues below are focused on additional information to include in the discussion. As written, the discussion sounds more like a summary of the results than a discussion of implications and relevant assumptions.

Major Issues

- Authors state on line 141 of the introduction that one of the aims of this study was to 'create a comprehensive sQTL resource', but it was not clearly stated how the research public could gain access to the sQTL resource.
- One point that was not included in the discussion is potential differences in sQTL analyses that use isoform ratios vs eQTL analyses done with expression levels of individual isoforms in both methodology and in interpretation.
- No discussion was included related to the dependency between sQTL tests of different isoforms of the same gene. Throughout, it was shown that the number of significant isoforms is approximately twice the number of significant genes.
- There was little discussion related to the potential impact of dramatic differences in the FHS and JHS populations including race, treatment for hypertension, high lipid levels, and diabetes.

Minor Comments

- A table that compared the source and processing of the RNA samples between the JHS and FHS would help clarify differences and similarities between studies. The description of the processing in the methods section was not easily comparable between the two studies.
- More details are needed about how transcripts are quantitated using STAR. For example, how were multi-mapping reads handled? What the max number of multiple alignments that were reported?
- It is curious that more cis pairs validated in FHS validation set than trans pairs. There was no mention of this in the discussion.

- Line 391 – it isn't clear why there would be more 'distinct sentinel cis-sQTL' than isoforms based on how a sentinel cis-sQTL was defined.
- Table 3 – if all cis-sQTL isoform pairs are replicated in this table, why do some have NA values for the JHS cohort?
- Line 418 – The point of this introductory sentence is unclear. How are they judging that an isoform is strongly regulated by genetic variants? The cis-eQTL-Gene is at the gene-level, not the isoform or transcript level?
- It would be informative to compare the enrichment of cis-sQTL in mQTL, pQTL, and GWAS to the enrichment of cis-eQTL in these groups.
- Line 516 – referring to wrong figure
- It would be helpful to be consistent on which strand you are using to derive reported genotype, eg line 505 refers to T/T, T/C, and C/C whereas figure refers to A/A, A/G, and G/G
- Line 532 – refers to wrong figure panel
- Lines 709-711 – the last sentence of the paragraph is unclear
- Perhaps an alternative type of graphic would make Figure 2 more impactful, e.g., violin plots by quantile. Too many points make the denser areas hard to interpret.
- The take home message of figure 3B is not clear. Too many overlapping points make it hard to decipher.
- Y-axis labels on Figure 5D would be helpful
- Labeling the location of the SNP on Figure 4A would be helpful

Version 2:

Reviewer comments:

Reviewer #1

(Remarks to the Author)

The authors performed an irQTL analysis of low-frequency variants, which further enhances the value of this paper. I have no further comments.

Reviewer #2

(Remarks to the Author)

The authors have sufficiently addressed my comments.

Reviewer #3

(Remarks to the Author)

The authors have done a thorough job of addressing the reviewer comments. No major/minor concerns remain.

Open Access This Peer Review File is licensed under a Creative Commons Attribution 4.0 International License, which permits use, sharing, adaptation, distribution and reproduction in any medium or format, as long as you give appropriate credit to the original author(s) and the source, provide a link to the Creative Commons license, and indicate if changes were

made.

Per Nature Communications manuscript NCOMMS-24-78200-T

Dear Reviewers,

We sincerely thank you for your thoughtful and constructive comments that have been invaluable for improving our manuscript. We have carefully addressed each point in the accompanying point-by-point response. In this reply, the reviewers' comments are italicized, followed by our detailed responses.

In the revised manuscript, we have highlighted the changes in yellow for ease of review. We have also included a clean version of the manuscript without highlights.

We believe that the revisions have significantly strengthened the manuscript and that it is now well-suited for publication in *Nature Communications*. We greatly appreciate your time and consideration.

Yours sincerely,

Chunyu Liu, PhD
Department of Biostatistics
Boston University School of Public Health

REVIEWER COMMENTS

Reviewer #1 (Remarks to the Author):

This study conducted a large-scale sQTL analysis on PBMC samples, identifying a substantial number of sQTL variants associated with isoform ratios. The manuscript is well-written, and the methodologies are generally applied appropriately. However, I believe there are several important concerns that need to be addressed by the authors to clearly distinguish the novelty of the findings.

1) I was surprised that the authors did not reference their previous sQTL study on the same FHS cohort, which utilized PBMC samples (Nat Genet 2015; doi: 10.1038/ng.3220). The authors should explicitly highlight the novel findings of the current study in comparison to their previous work. If this study represents a secondary analysis of the data, please clearly state this in the Methods section.

Reply: We thank the reviewer for referring to our previous paper, which was not cited in our current manuscript because this previous paper used different platforms and defined sQTLs differently, as described below.

1. Different platforms: Our previous paper (Nat Genet 2015) used gene expression data measured using the Affymetrix Exon Array and imputed single nucleotide variants (SNVs) based on 1000 Genomes Project Phase 1 data, using genotyping from the Affymetrix 500K Mapping Array (250K Nsp and 250K Sty arrays) and the Affymetrix 50K Supplemental Array. In contrast, our current manuscript uses gene expression data measured by RNA-seq and SNVs from whole genome sequencing (WGS). The older technologies used in that study are from 10 years ago and are inferior to today's RNA-seq and WGS platforms.

2. Different Definition of sQTLs: In our previous study, we conducted an association analysis between SNVs and exon-level expression (i.e., exon-level eQTL analysis). In that work, cis-sQTLs were defined as SNVs located within 50 kb of a gene boundary that showed a significant association ($P < 2.8e10$) with exon-level expression, specifically for genes without detectable gene-level expression associations. In the current study, we analyze associations between isoform ratios and SNVs located within 500 kb of a gene boundary.

In response to your comment, we have now included a citation, reference #21, to our prior paper in the revised Introduction section.

2) As suggested by the title, one of the strengths of this study appears to be the extensive whole-genome sequencing data. However, the authors did not fully capitalize on its potential benefits, as the analysis was limited to variants with a MAF > 0.01. The sQTL effects of rare variants could be assessed using alternative methodologies, such as evaluating allelic imbalance of isoform expression. If the number of variants is too large, silico filtering methods like SpliceAI or MaxENT could be employed.

Reply: Thank you for these comments. Based on the reviewer's suggestion, we have now included rare variants in the revised manuscript. We added the following in the revised manuscript, on page 8, line 20:

"Rare cis-sQTLs

As a secondary analysis, we assessed rare *cis*-sQTL variants (MAF 0.003 - 0.01). We identified 2,327 sentinel rare variants *cis*-sQTLs associated with 2,467 isoforms ($p < 5e-8$). This yielded 3,102 significant *cis*-sQTL-isoform pairs across 1,428 sGenes in the discovery sample (**Supplemental Table 4**). Among these sGenes, more than 94% had isoforms whose ratios were also associated with *cis*-sQTLs from common variants. Some rare variants explained a much larger R^2 of the isoform ratios than the lead common *cis*-sQTL for the same sGenes. For example, the rare variant rs1128271 (MAF=0.0093) explained approximately 38% of the variance in isoform ratios of ENST00000301788.12 in *POLR2G*, whereas the strongest common *cis*-sQTL, rs144760211 (MAF=0.03) explained 9.0% of the variance in the same isoform in the FHS discovery sample (**Supplemental Tables 2 & 4**).

On page 11, from line 1:

“Replication of rare cis-sQTLs

Among the 3,102 significant *cis*-sQTL-isoform pairs involving rare variants, replication rates increased with higher R^2 values of the *cis*-sQTLs (**Supplemental Tables 4 & 7, Supplemental Figure 4**). However, overall replication rates of rare variant *cis*-sQTL-isoform pairs remained substantially lower than those observed for common variants, only 22.2% in FHS and 40.7% in WHI. The lower replication rate in the additional FHS sample likely reflects its smaller sample size ($n = 1,094$) compared to WHI ($n = 2,005$). JHS ($n=1,010$) showed a markedly lower replication rate of just 9.4% (**Supplemental Tables 4 & 7, Supplemental Figure 4**). ”

3) *As discussed in the main text, the replication rate of sQTL signals was notably low at 23%. The authors speculated that this could be attributed to factors such as sample size, cellular population differences in PBMCs, and the distinct genetic ancestry of the replication samples. However, since the isoform ratio approach relies on the estimation of isoform expression using assembly tools, which may introduce biases, the possibility of false-positive findings cannot be ruled out. Therefore, I suggest that the authors validate the sQTL signals using junction-read-based approaches, such as LeafCutter.*

Reply: We thank the reviewer for this suggestion. In the revised manuscript, we have added an external replication analysis using the WHI sample with whole blood RNA-seq and WGS data. Additionally, we jointly normalized WHI, JHS, and FHS replication samples to ensure consistency across datasets.

Our external replication in WHI was robust with a replication rate of 72.4% for common variants and 40.7% for rare variants, supporting our earlier explanation that the low replication in JHS was primarily driven by cell type, specifically, RNA-seq from whole blood versus (in FHS and WHI) vs. PBMCs (in JHS).

In the revised manuscript, we added the following, on page 9, from line 15:

“Replication of common trans-sQTLs

The average replication rates of the 1,176 significant *trans*-sQTL-isoform pairs were 80.8% in FHS and 64.3% in WHI. Similarly, replication rates increased with R^2 values of the *trans*-sQTLs. For example, among sQTLs in the first R^2 quintile, replication rates were 58.0% in FHS and 37.0% in WHI, whereas in the fourth quintile, the rates rose to 94.0% and 79.6%, respectively (**Supplemental Tables 3 & 6, Supplemental Figure 3**).

Similar to what was observed for *cis*-sQTLs, the replication rate of *trans*-sQTL-isoform pairs was substantially lower in JHS (average rate = 17.5%) compared to WHI (average rate = 64.3%) (**Supplemental Tables 3 & 6, Supplemental Figure 3**).

Replication of rare cis-sQTLs

Among the 3,102 significant *cis*-sQTL-isoform pairs involving rare variants, replication rates increased with higher R^2 values of the *cis*-sQTLs (**Supplemental Tables 4 & 7, Supplemental Figure 4**). However, overall replication rates of rare variant *cis*-sQTL-isoform pairs remained substantially lower than those observed for common variants, only 22.2% in FHS and 40.7% in WHI. The lower replication rate in the additional FHS sample likely reflects its smaller sample size ($n = 1,094$) compared to WHI ($n = 2,005$). JHS ($n=1,010$) showed a markedly lower replication rate of just 9.4% (**Supplemental Tables 4 & 7, Supplemental Figure 4**).

We did not validate the sQTL signals using junction-read-based approaches such as LeafCutter, as our external replication rate was excellent once we introduced the WHI for external replication. Additionally, a key limitation of junction-based methods is their inability to reliably distinguish between isoforms that share similar or identical splice junctions, making it challenging to capture complex splicing events or quantify isoform-specific regulation. To overcome this limitation, we chose to analyze isoform ratios to identify sQTLs, an approach that was previously used by Yamaguchi et al. (reference 18). We added the following in the revised Discussion, page 21, from line 3:

“Unlike junction-based methods, our approach captures the ratio of isoform to total gene expression, potentially improving functional interpretation but with limitations for unknown isoforms.¹⁸”

4) *The sQTL effects on isoform ratios can also be validated using long-read sequencing data, such as those provided by the GTEx project (Nature 2022; doi: 10.1038/s41586-022-05035-y).*

Reply: We thank the reviewer for this comment. Obtaining long-read sequencing data for several thousand samples would require a substantial budget that is beyond the resources available for this project. Given the dramatic improvement in our replication rate by including WHI in the revised manuscript, we believe that our current approach is well supported. We agree with the reviewer that future studies using long-read sequencing technologies could further validate and expand upon our findings.

5) *Please discuss the potential reasons behind the observed enrichment of sQTL signals with pQTLs. Could it be that pQTL signals are influenced by the isoform ratio of genes? If the authors have any hypotheses regarding this, please provide specific examples or evidence to support them.*

Reply: The enrichment of sQTLs with pQTLs may be driven by several factors, including shared regulatory elements (e.g., promoters or enhancers), co-regulation of transcription and splicing, or the involvement of SNPs that influence both isoform expression and protein translation.

We are currently investigating the relationship between pQTLs and sQTLs in an ongoing project, with a separate manuscript in preparation. Given the ambitious scope of the current

study and the complexity of sQTL–pQTL interactions, we have removed the pQTL enrichment analysis from this manuscript and will address it in a dedicated paper.

Reviewer #2 (Remarks to the Author):

In this study, the authors performed isoform ratio QTL mapping on a whole blood dataset, validated the associations on an external dataset, and analyzed the results with respect to functional characteristics, other molecular QTLs, and GWAS variants.

There are major issues with the methodology and presentation of this study that call into question the novelty and quality of the analyses presented.

1. *There is a conflation of important concepts as demonstrated in the first sentence of the introduction: "Transcriptomic isoform variation generates different messenger RNAs (mRNAs) from the same pre-mRNA, ...". This is only true of alternative splicing; other forms of isoform variation, namely alternative TSS and alternative polyA sites, are not generated from the same pre-mRNA.*

Indeed, the presentation of the study and downstream analyses focus on alternative splicing, and the xQTLs are referred to as sQTLs, which they define as "splice variation quantitative trait locus". But the molecular phenotypes used are isoform ratios, which represent multiple types of variation including alternative splicing, alternative TSS, and alternative polyA sites, and there does not appear to have been any steps to narrow or process the isoforms so that they reflect alternative splicing specifically. It is therefore unclear to what degree the results of the splice-related analyses are supported by the "sQTL" data.

Reply: Thank you for pointing this out! In the revised Introduction, we have revised the manuscript accordingly and included the following (page 5, from line 2):

"Isoform variation results from several mechanisms including alternative splicing, alternative transcription start sites, and alternative polyadenylation (polyA) sites.¹⁻³ These mechanisms generate distinct messenger RNAs (mRNAs) from the same gene, expanding proteomic diversity beyond the ~20,000 protein-coding genes in the human genome.⁴"

2. *There is an existing study that performed "sQTL" mapping on the Framingham Heart Study dataset: (Nat Genet 2015; doi: 10.1038/ng.3220)*

Zhang, X., Joehanes, R., Chen, B. et al. Identification of common genetic variants controlling transcript isoform variation in human whole blood. Nat Genet 47, 345–352 (2015). <https://doi.org/10.1038/ng.3220>

It is strange that this study was not cited or discussed in the present study, and that makes it difficult to assess the novelty of the present study. From a cursory examination, they appear similar.

Reply: We thank the reviewer for referencing our previous paper, which used different (10-year-old) platforms and defined sQTLs differently, as explained in our response to the first comment from Reviewer 1. In response to your feedback, we have now included a citation, reference #21, to this paper in the revised Introduction.

3. *There is heavy emphasis on the number of variant-phenotype pairs with significant association. The number of such associations is nearly meaningless, since it is well understood that in xQTL studies such as this, the vast majority of variants in the associated pairs are not involved in regulation but rather are in LD with the causal variants. Similarly, the reported*

percentages of such pairs that were validated by the replication analysis is not meaningful, because even in different cohorts with different ancestries, variants in close proximity are likely to have some level of LD in both cohorts. Therefore, finding that high percentages of variant-phenotype pairs replicate does not indicate how many of the pairs actually have a real causal or functional relationship.

Reply: Thank you for the insightful comment. In the revised manuscript, we now report discovery and replication results, all based on sentinel QTLs, on page 7, from line 9, as follows:

“Transcriptome-wide identification of sQTLs in the FHS discovery sample

Common cis-sQTLs

We identified 14,056 significant *cis*-sQTL-isoform pairs ($p < 5e-8$) in the discovery sample, comprising 11,425 sentinel *cis*-sQTLs (MAF ≥ 0.05) associated with 10,883 isoforms across 4,971 sGenes (**Supplemental Table 2**). A *cis*-sQTL was defined as a variant located within 1 Mb of an isoform’s transcription start site (TSS), and a sentinel *cis*-sQTL is the variant most strongly (i.e., lowest p-value) associated with an isoform. An sGene refers to a gene with one or more significant sQTLs, while an sGene may have one or more isoforms (median: 5; interquartile range: 1-36) associated with sentinel sQTLs. An isoform may be associated with multiple sentinel *cis*-sQTLs if they are in perfect linkage disequilibrium (LD $r^2 = 1$). The top 25 *cis*-sQTLs-sGenes pairs are listed in **Table 2**. The most significant sentinel *cis*-sQTL, rs4841, was positively associated with the ratio of isoform ENST00000519690.1 of *RPS14* and explained 96% variance (R^2) of the ratio in the FHS discovery sample. *RPS14* encodes ribosomal protein 14, which plays a crucial role in ribosome biogenesis and protein synthesis as well as cell proliferation and nuclear stress.^{25,26}

Common trans-sQTLs

Only common SNPs were analyzed in trans-sQTL analysis, where a trans-sQTL was defined as being located more than 1 Mb away from the TSS of an isoform. For each isoform, the sentinel trans-sQTL was identified on each of the 23 chromosomes (22 autosome and X chromosome). This analysis yielded 2,999 significant trans-sQTLs-isoform pairs ($p < 1.5e-13$), involving 1,870 trans-sQTLs and 1,084 isoforms across 590 sGenes in the discover sample (Supplemental Table 3). Notably, the largest fraction of these pairs (64%) involved transcripts from genes on chromosome 6 that were significantly associated with sQTLs across the genome, with more than half of these genes located in the HLA (Human Leukocyte Antigen region)²⁷ region. Additionally, over 75% of these pairs involved sQTLs and gene transcripts located on different chromosomes (Supplemental Figure 1). The most significant of these inter-chromosomal pair was rs1458255–ENST00000418824, with the sQTL on chromosome 4 explained 54.2% variance in the transcript of the RPL9P9 gene, a ribosomal protein L9 pseudogene 9 gene, located on chromosome 15. RPL9P9 a non-functional copy of the RPL9 gene, which encodes a ribosomal protein that is a component of the 60S ribosomal subunit.²⁸

Rare cis-sQTLs

As a secondary analysis, we assessed rare *cis*-sQTLs variants (MAF 0.003 - 0.01) and identified 2,327 sentinel rare variants as *cis*-sQTLs associated with 2,467 isoforms ($p < 5e-8$). This yielded 3,102 significant *cis*-sQTL-isoform pairs across 1,428 sGenes in the discovery sample (**Supplemental Table 4**). Among these sGenes, more than 94% had isoforms whose ratios that were also associated with *cis*-sQTLs from common variants. Some rare variants explained a much larger R^2 of the isoform ratios than the lead

common *cis*-sQTL for the same sGenes. For example, the rare variant rs1128271 (MAF=0.0093) explained approximately 38% of the variance in isoform ratios of ENST00000301788.12 in *POLR2G*, whereas the strongest common *cis*-sQTL, rs144760211 (MAF=0.03) explained 9.0% of the variance in the same isoform in the FHS discovery sample (**Supplemental Tables 2 & 4**).

Replication of sQTLs

A significant QTL-isoform pair was considered replicated if it had a p-value < 1e-4 and showed the same direction of effect between the discovery and replication samples. The replication rate was defined as the proportion of sQTL-isoform pairs present in both the discovery and replication samples that met the replication criteria. We reported replication rates for sQTL-isoform pairs for sentinel sQTL.

Replication of common cis-sQTLs

Among the 14,056 significant *cis*-sQTL-isoform pairs ($p < 5e-8$) identified in the discovery sample, 10,783 (76.7%) were internally replicated in the additional FHS participants, and 10,174 (72.4%) were externally replicated in the WHI participants (Figure 2, Supplemental Table 5). R^2 represents the proportion of variance in an isoform ratio explained by its corresponding sQTL. Replication rates increased with higher R^2 values, as demonstrated across deciles of sentinel sQTL R^2 . For example, among sQTLs in the second decile, replication rates were approximately 45% in FHS and 37% in WHI, respectively, whereas in the eighth decile, these rates increased to 97% and 94% (Figure 2A). We also compared the effect sizes and directions of association for the 10,194 *cis*-sQTL-isoform pairs present in both the discovery and WHI replication samples. Approximately 99.8% of these pairs ($n = 10,174$) exhibited the same direction of association and consistent effect sizes between the two cohorts (Figure 2B, Supplemental Table 2).

Unlike FHS and WHI, where RNA-seq was performed on whole blood-derived RNA, JHS used RNA-seq from PBMCs (Supplemental Material), limiting direct comparability. As expected, the overall replication rate of 34.2% was much lower compared to that in WHI, although replication rates increased with higher R^2 values of the *cis*-sQTLs (Supplemental Tables 2 & 5, Supplemental Figure 2).

Replication of common trans-sQTLs

The average replication rates of the 1,176 significant *trans*-sQTL-isoform pairs were 80.8% in FHS and 64.3% in WHI. Similarly, replication rates increased with R^2 values of the *trans*-sQTLs. For example, among sQTLs in the first R^2 quintile, replication rates were 58.0% in FHS and 37.0% in WHI, whereas in the fourth quintile, the rates rose to 94.0% and 79.6%, respectively (Supplemental Tables 3 & 6, Supplemental Figure 3).

Similar to what was observed for *cis*-sQTLs, the replication rate of *trans*-sQTL-isoform pairs was substantially lower in JHS (average rate = 17.5%) compared to WHI (average rate = 64.3%) (Supplemental Tables 3 & 6, Supplemental Figure 3).

Replication of rare cis-sQTLs

Among the 3,102 significant *cis*-sQTL-isoform pairs involving rare variants, replication rates increased with higher R^2 values of the *cis*-sQTLs (Supplemental Tables 4 & 7, Supplemental Figure 4). However, overall replication rates of rare variant *cis*-sQTL-isoform pairs remained substantially lower than those observed for common variants, only 22.2% in FHS and 40.7% in WHI. The lower replication rate in the additional FHS

sample likely reflects its smaller sample size (n = 1,094) compared to WHI (n = 2,005). JHS (n=1,010) showed a markedly lower replication rate of just 9.4% (Supplemental Tables 4 & 7, Supplemental Figure 4)."

Other comments:

1. *Line 128: "They found that expression levels of nearly all expressed genes (eGenes) are subject to genetic regulation." The cited GTEx study defines 'eGenes' as "genes with an eQTL", not "expressed genes".*

Reply: Thank you for pointing this out. In the revised manuscript, we have modified the sentence as follows (on page 5, from line 18):

"They found that nearly all genes have at least one eQTL, highlighting the widespread influence of genetic variation on transcriptional regulation."

2. *The abstract mentions "allelic variation" twice in reference to a specific analysis, but doesn't define it, and the term is not used again in the text, so I can't tell if the analysis is actually presented. Perhaps it is just referring to the genotypic variation on which the QTLs are determined, but then this sentence: "We identified over 3.5 million cis-sQTL-isoform pairs ($p < 5e-8$), comprising 1,176,624 cis-sQTL variants and 10,883 isoform transcripts from 4,971 sGenes, with significant change in isoform-to-gene ratio due to allelic variation." would certainly be false, as the end of the sentence asserts a causal relationship, which is not the case for the vast majority of those 3.5 million cis-sQTL-isoform pairs.*

Reply: Thank you for this comment. After revising our manuscript, one sentence now includes the term "genotype variation", on page 4, from line 10:

"For 20% of cis-sQTLs in the FHS discovery sample, genotype variation was not significantly associated with overall gene expression, suggesting that it independently regulates isoform expression."

The reviewer is also correct for pointing out the confusion about causal relationships. A causal relationship is not appropriate here to describe *cis-sQTL-isoform pairs*. We have replaced "due to" with "associated with" in the Abstract. We have also updated our results to focus on sentinel sQTLs rather than all significant sQTLs. The original sentence is no longer relevant.

Reviewer #3 (Remarks to the Author):

This manuscript identifies splice QTL in blood to better understand the genetic influences on alternative splicing and alternative splicing as a potential molecular mechanism for how GWAS SNPs influence disease. The authors use isoform ratios as their quantitative measure of alternative splicing and include both a within sample validation (i.e., split the original cohort into a discovery and validation set of samples) and a cross-population validation (i.e., samples from a completely independent cohort with many differences in sample demographics and in RNA isolation and processing). This is a well-thought-out study with a tremendous amount of data and impactful results. In general, major issues below are focused on additional information to include in the discussion. As written, the discussion sounds more like a summary of the results than a discussion of implications and relevant assumptions.

Major Issues

1. Authors state on line 141 of the introduction that one of the aims of this study was to 'create a comprehensive sQTL resource', but it was not clearly stated how the research public could gain access to the sQTL resource.

Reply: Upon publication, the sQTL resource generated in this study will be available through the Gene Expression Omnibus (GEO), a public database maintained by the National Center for Biotechnology Information (NCBI).

2. One point that was not included in the discussion is potential differences in sQTL analyses that use isoform ratios vs eQTL analyses done with expression levels of individual isoforms in both methodology and in interpretation.

Reply: We thank the reviewer for raising this point. We have added the following sentences in the revised Discussion (page 21, from line 12):

“...using isoform ratios enables us to specifically capture splicing regulations independent of overall gene expression. A 'pure' eQTL may influence total gene expression without affecting isoform proportions, while a 'pure' sQTL alters isoform usage without changing total expression. In practice, we observed both scenarios, though most sQTLs in our study appear to be hybrid, affecting both isoform usage and overall expression. Importantly, while eQTLs linked to a specific isoform often reflect total gene expression changes, sQTLs may act independently of gene-level expression.”

3. No discussion was included related to the dependency between sQTL tests of different isoforms of the same gene. Throughout, it was shown that the number of significant isoforms is approximately twice the number of significant genes.

Reply: Thank you for your comments. In the revised Discussion, we added the following to address the dependency between sQTL tests of different isoforms of the same gene, on page 21, from line 3:

“In this study, we performed analyses based on isoform ratios. Unlike junction-based methods, our approach captures the ratio of isoform to total gene expression, potentially improving functional interpretation but with limitations for unknown isoforms. This isoform ratio method introduces interdependency among isoforms of the same gene, as their

proportions must sum to 1. Consequently, an increase in the expression ratio of one isoform may correspond to a decrease in one or more others. However, this dependency does not imply that all isoforms of a gene are regulated by the same variant. In some cases, such as *CNN2*, different isoforms are associated with distinct sentinel or secondary sQTLs, suggesting partially independent regulatory mechanisms.”

As noted by the reviewer, the number of significant isoforms is approximately twice the number of significant genes overall (e.g., 10,883 isoforms across 4,971 sGenes). Upon reviewing the data, we found that this appears to be coincidental. We summarized the significant isoforms (n=10,883) per gene for n=4,971 sGenes. Although each gene may have multiple isoforms, only one or a few may be identified as significant. For example, among the 4,971 sGenes, 2,087 have exactly one significant isoform, while in one case, a single sGene has 19 significant isoforms.

#significant isoforms	# sGene	Total
1	2087	2087
2	1529	3058
3	647	1941
4	310	1240
5	162	810
6	101	606
7	55	385
8	36	288
9	16	144
10	11	110
11	6	66
12	4	48
13	4	52
14	1	14
15	1	15
19	1	19
Total	4971	10883

4. *There was little discussion related to the potential impact of dramatic differences in the FHS and JHS populations including race, treatment for hypertension, high lipid levels, and diabetes.*

Reply: We thank the reviewer for this suggestion. In the revised manuscript, we have added an external replication analysis using the WHI sample with whole blood RNA-seq data. Additionally, we jointly normalized WHI, JHS, and FHS replication samples to ensure consistency across datasets.

Our external replication rate in WHI was 72.4%, supporting our earlier explanation that the lower replication in JHS was primarily driven by cell type, specifically, RNA-seq using whole blood in FHS and WHI versus PBMCs in JHS.

Please see our reply to the 3rd comment of Reviewer 2.

Minor Comments

1. A table that compared the source and processing of the RNA samples between the JHS and FHS would help clarify differences and similarities between studies. The description of the processing in the methods section was not easily comparable between the two studies.

Reply: See our reply to this reviewer's comment #4.

2. More details are needed about how transcripts are quantitated using STAR. For example, how were multi-mapping reads handled? What the max number of multiple alignments that were reported?

Reply: Thank you for this question. Many of the needed details are now included in the Supplementary Materials, including a list of parameters used when running STAR. The calls made by the STAR program are part of a standardized pipeline in TOPMed, a summary description is available at https://github.com/broadinstitute/gtex-pipeline/blob/master/TOPMed_RNAseq_pipeline.md. The STAR option switch – outFilterMultimapNmax defaults to a value of 20, meaning that reads are only output if they map to 20 or fewer locations.

3. It is curious that more *cis* pairs validated in FHS validation set than *trans* pairs. There was no mention of this in the discussion.

Reply: We think that the reviewer meant to ask why more *trans* pairs validated in the FHS replication set than *cis* pairs. Please see below from the previous submission.

“In the FHS discovery sample, we identified a total of 3,588,153 significant *cis*-sQTL-isoform pairs, comprising 1,176,624 distinct *cis*-sQTL variants and 10,883 distinct isoform transcripts from 4,971 distinct sGenes (**Table 2**). Of these, 61% of the *cis* pairs were validated in the FHS replication sample. We also identified 328,997 significant *trans*-sQTL-isoform pairs, consisting of 131,804 distinct *trans*-sQTL variants and 1,054 distinct isoform transcripts from 590 distinct sGenes in the FHS discovery sample. Compared to *cis*-pairs, a larger proportion of *trans*-sQTL-isoform pairs ($n=235,807$, 72%) were validated in the FHS replication sample (**Table 2**). The sentinel *cis*-sQTL and *trans*-sQTLs identified in FHS were presented in **Supplemental Tables 2 and 3**’

We believe this is due to consistent conditions and procedures used to process blood samples, extract RNA, conduct RNA-seq, and analyze the data. In the revised manuscript, we presented sentinel sQTLs, as suggested by the reviewers. We found a consistent trend for sentinel *trans*-sQTLs as we did for all significant pairs in our previous version. However, in the WHI external replication, the replication rate was higher for *cis* than *trans* pairs. In the revised Discussion, on page 22, from line 1, we added the following to discuss our findings:

“Using this method, the external replication rate was higher for *cis*-sQTLs than *trans*-sQTLs, which is consistent with previously reported findings.^{55,56} This is likely due to the genetic proximity of *cis*-sQTLs to target isoforms and their involvement in local regulatory mechanisms. Despite this, we observed more replicated *trans*-sQTL-isoform pairs than *cis* pairs in the additional FHS sample. This may be attributed to the consistent experimental conditions and analytical pipelines used across FHS discovery

and replication samples, including sample processing, RNA extraction, sequencing, and data analysis.”

4. *Line 391 – it isn’t clear why there would be more ‘distinct sentinel cis-sQTL’ than isoforms based on how a sentinel cis-sQTL was defined.*

Reply: More than one *cis*-SNP in perfect linkage disequilibrium (LD, $r^2 = 1$) may be associated with the same isoform, resulting in more sentinel *cis*-sQTLs than isoforms.

We added this missing explanation to the revised Results on page 7, from line 18:

“An isoform may be associated with multiple sentinel *cis*-sQTLs if they are in perfect linkage disequilibrium (LD $r^2 = 1$).”

5. *Table 3 – if all cis-sQTL isoform pairs are replicated in this table, why do some have NA values for the JHS cohort?*

Reply: Table 3 in the previous version of the manuscript is now Table 2. We added WHI as our external replication cohort, and the revised Table 2 no longer contains any 'NA' entries.

6. *Line 418 – The point of this introductory sentence is unclear. How are they judging that an isoform is strongly regulated by genetic variants? The cis-eQTL-Gene is at the gene-level, not the isoform or transcript level?*

Reply: We agree with your comment. We revised the manuscript accordingly. That introductory sentence is no longer relevant.

7. *It would be informative to compare the enrichment of cis-sQTL in mQTL, pQTL, and GWAS to the enrichment of cis-eQTL in these groups.*

Reply: We thank the reviewer for this insightful comment.

We are currently investigating the relationship between pQTLs or mQTLs and sQTLs in two ongoing projects, with separate manuscripts in preparation. Given the ambitious scope of the current study and the intricate nature of sQTL-pQTL and sQTL-mQTL relationships, we have chosen to remove the enrichment analyses from this manuscript. These analyses will be presented in greater depth in forthcoming dedicated papers.

8. *Line 516 – referring to wrong figure.*

Reply: In the revised manuscript, we carefully reviewed the cited Figure numbers and corrected all typos.

9. *It would be helpful to be consistent on which strand you are using to derive reported genotype, eg line 505 refers to T/T, T/C, and C/C whereas figure refers to A/A, A/G, and G/G*

Reply: Thank you for the suggestion. We have revised the manuscript to consistently use the same strand for reporting genotypes across the text and figures.

In the revised manuscript, page 15, from line 4, we added the following clarification:

“Gene transcription occurs in the 5' to 3' direction. Since *ULK3* is on the negative strand, we converted the nucleotide sequence from the positive strand to its complementary sequence on the negative strand around rs12898397 while preserving the 5' to 3' orientation. The complementary nucleotide sequence (e.g., for rs12898397, T>C → A>G) was used for analysis.”

In the revised Figure 5, we revised the legend as follows:

“**Figure 5A.** *ULK3* (unc-51 like kinase 3) lies on the negative strand of chromosome 15 in the hg38 genome build of the human genome. It has 16 exons, with the sentinel variant rs12898397 (T>C) located within the 14th exon. Gene transcription occurs in the 5' to 3' direction. Since *ULK3* is on the negative strand, we converted the nucleotide sequence from the positive strand to its complementary sequence on the negative strand around rs12898397, while preserving the 5' to 3' orientation. The complementary nucleotide sequence (e.g., for rs12898397, T>C → A>G) was used for analysis.”

10. Line 532 – refers to the wrong figure panel

Reply: In the revised manuscript, we corrected the typo.

11. Lines 709-711 – the last sentence of the paragraph is unclear.

Reply: In the revised manuscript, we added analysis of rare variants ($0.003 < \text{MAF} < 0.01$). The last sentence is no longer relevant.

12. Perhaps an alternative type of graphic would make Figure 2 more impactful, e.g., violin plots by quantile. Too many points make the denser areas hard to interpret.

Reply: Due to our revised replication analysis, we updated Figure 2 with its legend as shown below.

Figure 2. Replication of sentinel *cis*-sQTLs in relation to their associated isoforms. We identified 11,425 common variants ($MAF \geq 0.05$) as sentinel *cis*-sQTLs for 10,883 isoforms, comprising 4,971 sGenes in the discovery FHS sample ($n=2,622$). Replication was performed in an independent FHS sample ($n=1,094$), and in WHI sample ($n=2,005$). An association of a sentinel *cis*-sQTL with its isoform ratio was considered replicated if its $p < 1e-4$ and the effect direction was consistent with discovery. (A) Replication rates of the sentinel *cis*-sQTL increases with R^2 , the proportion of variance of the corresponding isoform ratio explained by the sQTLs. Average replication rates were 76.7% and 72.4%, respectively. (B) Comparison of the sentinel *cis*-sQTL-isoform ratio effects between the FHS discovery sample and WHI. Among all *cis*-sQTL-isoform pairs present in both FHS discovery and WHI replication, 99.8% showed the same direction of association.

13. The take home message of figure 3B is not clear. Too many overlapping points make it hard to decipher.

Reply: We revised Figure 3B (see below) and revised the legend to provide a clearer take-home message.

Figure 3B. Comparison of strength of association, R^2 or proportion of variance of isoform ratio or gene expression explained by genotype of *cis*-eQTLs for 4,971 sGenes. Three regions are defined by dashed lines, including 1946 SNPs with stronger evidence for an sQTL (cyan color), 2716 SNPs with evidence for both an sQTL and eQTL (middle region), and the remaining SNPs with stronger evidence for eQTL behavior (red color). Three highlighted genes, *OAS1*, *ULK3* and *CNN2* were analyzed in detail and reported in the main text.

14. Y-axis labels on Figure 5D would be helpful.

Reply: We added the Y-axis label in the revised Figure (Now Figure 5B).

15. Labeling the location of the SNP on Figure 4A would be helpful.

Reply: We added the location of the SNP on Figure 4A. We also modified the legend for 4A.

Figure A. *OAS1* lies on the long arm of chromosome 12 on the positive strand in the hg38 genome build of the human genome and its expression product (pre-mRNA) is subjected to editing. The rs10774671 (G>A) lies at the splice acceptor site of intron 5

Dear Reviewers,

We sincerely thank you for your thoughtful and constructive comments that have been invaluable for improving our manuscript. We have carefully addressed each point in the accompanying point-by-point response. In this reply, the reviewers' comments are italicized, followed by our detailed responses.

In the revised manuscript, we have highlighted the changes in yellow for ease of review. Below we summarize key changes in the revised manuscript:

1. Compared key differences between the 2015 paper (Zhang et al., Nature Genetics) and our current paper.
2. Changed sQTLs to irQTLs throughout the manuscript.
3. Added statistical analysis of variants with $0.003 \leq \text{MAF} < 0.01$.
4. Added additional replication cohort, Women's Health Initiative (WHI), with WGS and RNA-seq data from whole blood. We conducted analyses in the full sample, as well as separately in the White American and African American subgroups, to investigate whether the low replication rate observed in JHS is more likely attributable to RNA source, race, or clinical characteristics.
5. Added multiple layers of annotation to the discovered irQTLs using a range of databases, including GENCODE, SpliceAI, SnpEff, and HOMER, to provide insights into their potential functional and regulatory roles.
6. Added text in Discussion to respond to Reviewer Comments.

We believe that the revisions have significantly strengthened the manuscript and that it is now well-suited for publication in *Nature Communications*. We greatly appreciate your time and consideration.

Yours sincerely,

Chunyu Liu, PhD
Department of Biostatistics
Boston University School of Public Health

REVIEWER COMMENTS

Reviewer #1 (Remarks to the Author):

This study conducted a large-scale sQTL analysis on PBMC samples, identifying a substantial number of sQTL variants associated with isoform ratios. The manuscript is well-written, and the methodologies are generally applied appropriately. However, I believe there are several important concerns that need to be addressed by the authors to clearly distinguish the novelty of the findings.

1) I was surprised that the authors did not reference their previous sQTL study on the same FHS cohort, which utilized PBMC samples (Nat Genet 2015; doi: 10.1038/ng.3220). The authors should explicitly highlight the novel findings of the current study in comparison to their previous work. If this study represents a secondary analysis of the data, please clearly state this in the Methods section.

Reply: We appreciate the opportunity to address this important point. Respectfully, we would like to clarify that our study is not merely a secondary analysis of the 2015 data, as suggested. Rather, it represents a substantially expanded investigation of completely original data. The key differences are summarized in **Supplemental Table 1**.

In the revised Introduction, we added the following paragraph to explicitly state the distinction of our present work from the 2015 study. On page 4, from line 86:

“Comprehensive analyses of SNP associations with isoform variation in whole blood and their relevance to human health remain limited. In prior work, we identified exon-level expression variants in 2,650 genes using imputed SNPs and hybridization-based exon array data from 5,257 Framingham Heart Study (FHS) participants.²¹ Here, we present a new study leveraging WGS and RNA-seq data from 3,716 FHS participants (2,622 for discovery, 1,094 for internal replication) to map isoform ratio QTLs (irQTLs) as described originally by Yamaguchi et al. (2022)¹⁸ (**Supplemental Table 1**). We further evaluate these findings in two external cohorts with WGS and RNA-seq data.^{23,24}”

In the revised Results, on page 9, from line 207, we added:

“A prior splicing QTL study in FHS identified exon-level QTLs in 2,650 genes using imputed SNPs and exon array data.²¹ That study also reported the most significant splicing QTLs within 50 kb of each gene. Only 458 (17.3%) of gene–SNP pairs from the earlier study showed $P < 1 \times 10^{-4}$ in our discovery sample (**Supplemental Table 8**). This modest replication rate likely reflects differences in technology (imputed SNPs vs. WGS; exon arrays vs. RNA-seq) and methodology (SNPs associated with exon expression vs. isoform ratio changes) between the earlier and current studies.”

We added the following text in revised Discussion, page 20, from line 455:

“Our study confirmed a subset of previously reported splicing QTL–gene pairs from FHS,²¹ identifying concordant associations for 17.3% of the 2,650 pairs tested ($P < 1 \times 10^{-4}$). The limited overlap may reflect differences in study design and methodology, including genetic and transcriptomic platforms (imputed SNPs vs. WGS; exon arrays vs. RNA-seq) and the expression features tested (exon-level expression vs. isoform ratios). Some previously reported associations may also represent false positives, highlighting the importance of independent validation with high-resolution data.”

Supplemental Table 1. Comparison of Zhang et al. (2015) paper with our current study

Characteristics	2015 study	Current study
Study population	Framingham Heart Study	Framingham Heart Study
Discovery n	n=5257	n=2662
SNP identification	Imputation based on SNP array data aligned to hg19	SNPs from whole-genome sequencing aligned to hg38
Number of SNPs	9,883,769	12,887,093
RNA platform	Affymetrix GeneChip Human Exon 1.0 ST aligned to hg19	Whole transcriptome sequencing aligned to hg38
Cis-window definition	± 50 kb from gene boundary	± 1Mb from gene transcription start site (TSS)
Trans-QTL	Not studied	Beyond ± 1Mb from TSS or on the different chromosome
Uncommon/rare QTLs	No	0.003<MAF<0.01 for cis-irQTL
QTL identification method	SNPs associated with exon expression	SNPs associated with isoform ratio
Replication	Findings from the 2015 study were not tested for replication in an independent dataset. The study did replicate some previously reported sQTL–gene pairs from Battle et al. (2014), which used RNA-seq and SNP imputation in 922 individuals.	Internal replication with additional 1094 samples
		External replication in Women's Health Initiative (n=2005) and Jackson Heart Study (n=1010)
Investigation of relationships between irQTLs, alternative splicing, and diseases/traits	No	Provide three examples to investigate the relationships.

On page 18, line 405, we compared the three methods in identifying irQTL, QTL for exon expression and QTL from LeafCutter:

“The isoform ratio method¹⁸ captures the relative expression of each isoform within a gene, allowing the detection of regulatory variation that affects full-length splice isoforms. It is particularly useful for identifying biologically interpretable changes, such as isoform switches that impact coding potential, untranslated regions, or transcript stability. Compared to exon-level QTLs,²¹ which detect local events like exon skipping, and junction-based LeafCutter QTLs,^{16,17} which are optimized for discovering novel or cryptic intron excision, the isoform ratio method provides a clearer view of transcript-level consequences (**Supplemental Table 16**). However, it relies on comprehensive and accurate transcript annotations, potentially missing events in poorly annotated genes. It also has lower resolution to identify specific splice sites or exons, may overlook subtle or complex splicing changes. Despite these limitations, its high biological interpretability and relevance to phenotype-driven studies make it a valuable complement to other QTL approaches.”

Supplemental Table 16. Compare methods: isoform ratio, exon-level QTL, and LeafCutter QTL

Method	Splicing resolution	Annotation dependency	Biological interpretability	Appropriate for
Isoform ratio QTL	Low (full transcript)	High	High (e.g., protein isoforms)	Transcript switches
Exon-level QTL	Medium (individual exons)	Medium	Medium	Exon skipping/inclusion
LeafCutter QTL	High (intron junctions)	Low	Low-to-medium	Novel or cryptic splice

2) As suggested by the title, one of the strengths of this study appears to be the extensive whole-genome sequencing data. However, the authors did not fully capitalize on its potential benefits, as the analysis was limited to variants with a MAF > 0.01. The sQTL effects of rare variants could be assessed using alternative methodologies, such as evaluating allelic imbalance of isoform expression. If the number of variants is too large, silico filtering methods like *SpliceAI* or *MaxENT* could be employed.

Reply: We thank the Reviewer for all suggestions.

Allelic imbalance is a useful approach for analyzing common variants, provided there is a sufficient number of heterozygous individuals and adequate read depth at the variant site. However, it is less effective for rare variants reflecting the scarcity of heterozygous individuals, even in large cohorts. Low read depth at rare variant sites further increases technical noise and biases, such as mapping errors and reference bias, making this strategy unreliable. (Mohammadi et al., *Genome Research*. 2017. 27(11):1872-1884. PMID: PMC5668944; Knowles et al., *Nat Methods*. 2017 Jun;14(6):590-592 PMID: PMC5482724).

When analyzing individual variants, a commonly used guideline is to include only those with a minor allele count (MAC) >20 in cohorts of approximately 1,000 individuals. This corresponds to a minor allele frequency (MAF) of about 1% and provides sufficient power for robust association testing and reliable effect estimation. We analyzed rare/uncommon variants with MAF >0.003, corresponding to >16 minor allele counts (MAC) in our discovery sample (n = 2,622). These thresholds represent the minimum required to ensure reliable analysis across cohorts.

Our group is currently conducting analyses of rare variants using aggregation methods such as SKAT with sliding window approaches, which represent a separate effort and manuscript.

We added the following in the revised Results, on page 7, line 162:

“Rare cis-irQTLs

We assessed *cis-irQTL* variants ($0.003 < \text{MAF} < 0.01$) in the discovery sample. This analysis identified 2,327 rare *cis-irQTL* variants associated with 2,467 isoforms ($P < 5 \times 10^{-8}$), yielding 3,102 significant *cis-irQTL*-isoform pairs across 1,428 genes (**Supplemental Table 5**). Among these genes, 6% had isoforms that were not associated with common variants. Of note, some rare variants explained a larger R^2 of the corresponding isoform ratios than any of the common *cis-irQTL* variant for the same gene. For example, the rare variant rs1128271 (MAF = 0.0093) explained approximately 38% of the variance in isoform ratios of ENST00000301788.12 in *POLR2G* --much more than the strongest more common *cis-irQTL*, rs144760211 (MAF = 0.03), which explained just 9% of the variance in the isoform ratio (**Supplemental Tables 3 & 5**). Twenty-eight

sentinel variants were annotated as splicing-related by SnpEff, and seven of them had a SpliceAI score greater than 0.2 (**Supplemental Table 5**)”

We added SpliceAI annotation to these rare variants in Supplemental Table 5.

On page 10, line 221, we added:

“Replication of rare cis-irQTLs

Among the 3,102 significant *cis*-irQTL-isoform pairs involving rare variants, replication rates also increased with higher R^2 values of the *cis*-irQTLs (**Supplemental Tables 5 & 9, Supplemental Figure 4**). However, overall replication rates of rare variant *cis*-irQTL-isoform pairs remained substantially lower than those observed for common variants, only 22.2% in the additional 1,094 FHS sample and 40.7% in WHI. The lower replication rate in FHS likely reflects its smaller sample size ($n = 1,094$) compared to WHI ($n = 2,005$). JHS ($n = 1,010$) showed a markedly lower replication rate of just 9.4%.”

In the Revised Discussion, page 20, line 462, we added:

“Our study has limited power to detect rare irQTLs reflecting constraints related to sample size and allele frequency thresholds. Larger and more diverse cohorts, along with alternative methods (e.g., SKAT⁶⁰ and SKAT-O⁶¹), will be needed to capture the full spectrum of rare splicing regulatory variation.”

3) *As discussed in the main text, the replication rate of sQTL signals was notably low at 23%. The authors speculated that this could be attributed to factors such as sample size, cellular population differences in PBMCs, and the distinct genetic ancestry of the replication samples. However, since the isoform ratio approach relies on the estimation of isoform expression using assembly tools, which may introduce biases, the possibility of false-positive findings cannot be ruled out. Therefore, I suggest that the authors validate the sQTL signals using junction-read-based approaches, such as LeafCutter.*

Reply: We thank the reviewer for this suggestion. In the revised manuscript, we have added an external replication analysis using the Women’s Health Initiative (WHI), which includes whole blood RNA-seq and WGS data, providing a comparable RNA source to that used in FHS. We jointly normalized WHI, JHS, and FHS replication samples to ensure consistency across datasets.

In the revised Abstract, we added the following text, on page 2, line 39:

“We developed a comprehensive whole-blood isoform ratio QTL (irQTL) resource by analyzing genome-wide isoform-to-gene expression ratios using whole-genome and transcriptome sequencing data. In the Framingham Heart Study (FHS discovery set, $n = 2,622$), we identified over 1.1 million *cis*-irQTLs ($MAF \geq 0.01$, within ± 1 Mb of 10,883 isoform transcripts, $P < 5 \times 10^{-8}$) across 4,971 genes. The replication rate ($P < 1 \times 10^{-4}$) for 11,425 most significant (i.e., sentinel) *cis*-irQTLs per isoform was 77% in a separate FHS sample ($n = 1,094$) and 72% in the Women’s Health Initiative (WHI external replication set; $n = 2,005$). Notably, 20% of these *cis*-irQTLs showed no significant association with overall gene expression, indicating isoform-specific effects. These variants were significantly enriched ($P < 1 \times 10^{-10}$) at splice donor/acceptor sites and GWAS loci. We also identified 1,870 sentinel *trans*-irQTLs ($P < 1.5 \times 10^{-13}$; $MAF \geq 0.01$), located beyond 1 Mb or on different chromosomes, associated with 1,084 isoforms across 590 genes, with similar replication rates. Additionally, we identified 2,327 rare *cis*-

irQTLs ($0.003 < \text{MAF} < 0.01$, $P < 5 \times 10^{-8}$) for 2,467 isoforms of 1,428 genes, 22% of which replicated internally and 41% replicated externally in WHI.”

In the revised Introduction, we added the following text, on page 4, line 93:

“We further evaluate these findings in two external cohorts with WGS and RNA-seq data.^{23,24}”

In the revised manuscript, we added the following, on page 8, from line 8:

“Replication of *cis*-irQTLs with $\text{MAF} \geq 0.01$

Among the 14,056 discovered sentinel *cis*-irQTL-isoform pairs ($P < 5 \times 10^{-8}$), 10,783 (76.7%) were internally replicated in an additional 1,094 FHS samples, and 10,174 (72.4%) were externally replicated in the WHI (**Figure 2, Supplemental Table 6**). R^2 represents the proportion of variance in an isoform ratio explained by its corresponding irQTL. Replication rates increased with higher R^2 values, as demonstrated in Figure 2A. For example, among irQTLs in the second decile, replication rates were approximately 46% in FHS and 44% in WHI, whereas in the eighth decile, these rates increased to 98% and 93%, respectively (**Figure 2A**). We also compared the effect sizes and directions of association for the 10,194 *cis*-irQTL-isoform pairs present in both the discovery and WHI replication samples. Approximately 99.8% of these pairs exhibited the same direction of association and highly concordant effect sizes (Pearson coefficient $r = 0.91$) (**Figure 2B, Supplemental Table 3**).

The overall replication rate of *cis*-irQTLs with $\text{MAF} \geq 0.01$ in JHS ($n = 1,010$) was 32.4%, substantially lower than in WHI, though replication improved with increasing *cis*-irQTL R^2 values (**Supplemental Tables 3 & 6, Supplemental Figure 2**). JHS isolated RNA from PBMCs, whereas FHS and WHI used whole blood-derived RNA, limiting direct comparability (**Supplemental Table 7**).²³ To further evaluate the effects of RNA source, ancestry, and sample size, separate analyses were conducted in African American ($n = 918$) and White American ($n = 1,000$) participants in WHI. Replication rates were lower in the White American sub-sample (60.2%), reflecting its smaller sample size, and even lower in the similarly sized African American sub-sample (49.0%), reflecting the potential influence of ancestry. This analysis suggests that the reduction in replication rate observed in the JHS cohort (32.4%) may largely reflect the use of a different RNA source.

A prior splicing QTL study in FHS identified exon-level QTLs in 2,650 genes using imputed SNPs and exon array data.²¹ That study also reported the most significant splicing QTLs within 50 kb of each gene. Only 458 (17.3%) of gene-SNP pairs from the earlier study showed $P < 1 \times 10^{-4}$ in our discovery sample (**Supplemental Table 8**). This modest replication rate likely reflects differences in technology (imputed SNPs vs. WGS; exon arrays vs. RNA-seq) and methodology (SNPs associated with exon expression vs. isoform ratio changes) between the earlier and current studies.

Replication of *trans*-irQTLs with $\text{MAF} \geq 0.01$

The average replication rates of the 2,999 sentinel *trans*-irQTL-isoform pairs were 80.8% in FHS and 60.8% in WHI. Similarly, replication rates increased with R^2 values of the *trans*-irQTLs (**Supplemental Tables 4 & 9, Supplemental Figure 3**). The replication rate of *trans*-irQTL-isoform pairs was again substantially lower in JHS (average rate = 20.2%) compared to WHI (average rate = 64.3%).

Replication of rare *cis*-irQTLs

Among the 3,102 significant *cis*-irQTL-isoform pairs involving rare variants, replication rates also increased with higher R² values of the *cis*-irQTLs (**Supplemental Tables 5 & 10, Supplemental Figure 4**). However, overall replication rates of rare variant *cis*-irQTL-isoform pairs remained substantially lower than those observed for common variants, only 22.2% in the additional 1,094 FHS sample and 40.7% in WHI. The lower replication rate in FHS likely reflects its smaller sample size (n = 1,094) compared to WHI (n = 2,005). JHS (n = 1,010) showed a markedly lower replication rate of just 9.4%.”

In the revised Discussion, we added the following, on page 19, line 435:

“In addition to FHS, we analyzed community-based cohorts, JHS²³ and WHI²⁴, to enable independent replication and enhance the generalizability of our findings across diverse populations. We observed a 72% replication rate in WHI, supporting the robustness of our results. In contrast, replication in JHS was lower (32%), likely reflecting both technical and biological differences. Notably, JHS used RNA-seq from PBMCs, while FHS and WHI used whole blood. To further assess the impact of tissue source and ancestry, we stratified WHI by ancestry. Replication was highest in the full WHI cohort and lower in ancestry-specific subsets, partly reflecting reduced sample size (**Supplemental Figure 2**). Despite similar sample sizes, replication in JHS African American participants was lower than in WHI African American participants, suggesting that tissue source may have a greater effect on replication than ancestry. While clinical characteristics also varied across cohorts (**Table 1, Supplemental Table 2**), their influence on splicing could not be directly assessed. These results underscore the importance of considering tissue type, ancestry, and sample size in cross-cohort transcriptomic analyses.”

We understand the strengths and limitations of both LeafCutter and the irQTL method. This has been addressed in the revised Discussion, on page 18, line 405:

“The isoform ratio method¹⁸ captures the relative expression of each isoform within a gene, allowing the detection of regulatory variation that affects full-length splice isoforms. It is particularly useful for identifying biologically interpretable changes, such as isoform switches that impact coding potential, untranslated regions, or transcript stability. Compared to exon-level QTLs,²¹ which detect local events like exon skipping, and junction-based LeafCutter QTLs,^{16,17} which are optimized for discovering novel or cryptic intron excision, the isoform ratio method provides a clearer view of transcript-level consequences (**Supplemental Table 16**). However, it relies on comprehensive and accurate transcript annotations, potentially missing events in poorly annotated genes. It also has lower resolution to identify specific splice sites or exons, may overlook subtle or complex splicing changes. Despite these limitations, its high biological interpretability and relevance to phenotype-driven studies make it a valuable complement to other QTL approaches.”

Supplemental Table 16. Compare methods: isoform ratio, exon-level QTL, and LeafCutter QTL

Method	Splicing resolution	Annotation dependency	Biological interpretability	Appropriate for
Isoform ratio QTL	Low (full transcript)	High	High (e.g., protein isoforms)	Transcript switches
Exon-level QTL	Medium (individual exons)	Medium	Medium	Exon skipping/inclusion

LeafCutter QTL	High (intron junctions)	Low	Low-to-medium	Novel or cryptic splice
----------------	----------------------------	-----	---------------	----------------------------

4) *The sQTL effects on isoform ratios can also be validated using long-read sequencing data, such as those provided by the GTEx project (Nature 2022; doi: 10.1038/s41586-022-05035-y).*

Reply: Glinos et al. performed long-read RNA sequencing using the Oxford Nanopore Technologies platform on 88 samples from GTEx tissues and cell lines (Nature. 2022 Aug;608(7922):353-359.)

The Glinos paper includes a supplementary table (Supplementary Table 10) listing 26 rare heterozygous variants associated with splicing outliers, based on long-read sequencing data. Of the 10 genes linked to these variants, 8 were also found to be associated with irQTLs in our analyses. Notably, one of Glinos' rare variants (chr17_81678961_G_C_b38), associated with the *ARL16* gene, was also identified in our study, but as a more common variant, rs112947993 (MAF = 0.012), at the same genomic position. The irQTL-isoform pair, rs112947993 - ENST00000571082.5 gave negative log₁₀ p-value=10.3 in our study. However, a nearby SNP, rs6565619 (EAF=0.233) located about 100 bp further downstream gave a much stronger negative log₁₀ pvalue of 154.4. Both Glinos' rare variant and our nearby much stronger more common variant were located downstream of the gene body. This suggests that our study is capable of detecting associations with variants that appear rare in smaller studies.

5) *Please discuss the potential reasons behind the observed enrichment of sQTL signals with pQTLs. Could it be that pQTL signals are influenced by the isoform ratio of genes? If the authors have any hypotheses regarding this, please provide specific examples or evidence to support them.*

Reply: The enrichment of irQTLs with pQTLs may be driven by several factors, including shared regulatory elements (e.g., promoters or enhancers), co-regulation of transcription and splicing, or the involvement of SNPs that influence both isoform expression and protein translation.

We are currently investigating the relationship between pQTLs and irQTLs in an ongoing project, with a separate manuscript in preparation. Given the ambitious scope of the current study and the complexity of irQTL–pQTL interactions, we have removed the pQTL enrichment analysis from this manuscript and will address it in a dedicated paper.

Reviewer #2 (Remarks to the Author):

In this study, the authors performed isoform ratio QTL mapping on a whole blood dataset, validated the associations on an external dataset, and analyzed the results with respect to functional characteristics, other molecular QTLs, and GWAS variants.

There are major issues with the methodology and presentation of this study that call into question the novelty and quality of the analyses presented.

Reply: We sincerely thank the Reviewer for their thoughtful feedback. We would like to clarify that our study is an independent and original investigation, utilizing comprehensive irQTL mapping through the integration of whole-genome sequencing and RNA-seq data. In addition to the internal and external replication included in our original submission, we have now conducted further analyses in response to the Reviewer's comments. Detailed responses are provided below, and we hope that these clarifications and new results clearly convey the rigor and novelty of our work.

1. *There is a conflation of important concepts as demonstrated in the first sentence of the introduction: "Transcriptomic isoform variation generates different messenger RNAs (mRNAs) from the same pre-mRNA, ...". This is only true of alternative splicing; other forms of isoform variation, namely alternative TSS and alternative polyA sites, are not generated from the same pre-mRNA.*

Indeed, the presentation of the study and downstream analyses focus on alternative splicing, and the xQTLs are referred to as sQTLs, which they define as "splice variation quantitative trait locus". But the molecular phenotypes used are isoform ratios, which represent multiple types of variation including alternative splicing, alternative TSS, and alternative polyA sites, and there does not appear to have been any steps to narrow or process the isoforms so that they reflect alternative splicing specifically. It is therefore unclear to what degree the results of the splice-related analyses are supported by the "sQTL" data.

Reply: Thank you for pointing this out. In response, we have revised the Introduction and incorporated the following changes, on page 3, line 62:

"Isoform variation results from several mechanisms including alternative splicing, alternative transcription start sites, and alternative polyadenylation (polyA) sites.¹⁻³ These mechanisms generate distinct messenger RNAs (mRNAs) from the same gene, expanding proteomic diversity beyond the ~20,000 protein-coding genes in the human genome.⁴ Alternative splicing, which joins exons in different combinations to produce mature RNA isoforms,⁵ can be both a cause and consequence of disease.^{6,7} Recent advances in sequencing technologies have revealed that up to 95% of multi-exon genes undergo alternative splicing in humans."

To address the Reviewer's comment regarding the use of sQTL, we have replaced the term "sQTL" with the more appropriate "isoform-ratio QTL (irQTL)" throughout the revised manuscript. We have also made the following modifications to address the reviewer's concerns (revised Results). on page 5, line 127:

"The irQTL method, introduced by Yamaguchi et al. (2022),¹⁸ may identify QTLs that affect splicing, as well as QTLs associated with changes in isoform ratios resulting from other regulatory mechanisms. To better interpret the molecular consequences of each variant, we integrated multiple annotation layers from the following databases: GENCODE,²⁹ SpliceAI,³⁰ SNPEff,³¹ and Homer.³² Each database offers distinct strengths and limitations, capturing different aspects of variant annotation. SnpEff³¹ is a variant

annotation tool that primarily uses Ensembl-based gene models, but it also supports RefSeq, GENCODE, UCSC Genome Browser annotations. According to SnpEff, 279 variants are related to splicing (e.g., as splice acceptor, splice donor, or splice region variants). SpliceAI,³⁰ a tool for predicting splice site alterations from genetic variants, predicted that 215 variants had SpliceAI prediction scores > 0.2, suggesting potential splice-altering effects, and 82 of these had SpliceAI prediction scores > 0.5, indicating a higher likelihood of splicing disruption. A total of 86 variants were identified by both SnpEff and SpliceAI as potentially splice-altering (**Supplemental Tables 3**).”

The following examples (see Reviewers Table below – reflecting information from Supplemental Table 3) illustrates the use of these annotations. The irQTL rs10774671 in *OAS1* is associated with three transcript isoforms, *OAS1-201*, *OAS1-130*, and *OAS1-364*, producing proteins of 400, 130, and 364 amino acids, respectively. Along with SpliceAI prediction, this SNP is clearly a splice altering variant. Similarly, rs73191241 appears to be a splice regulator for *SELPLG*, affecting alternative splicing of *SELPLG-202* and *SELPLG-203*, which encode proteins of 402 and 412 amino acids, respectively. However, SpliceAI did not provide a score for this variant.

SNP	Chr	Isoform ID	Gene	isoform	ProteinID	ProteinLen	SpliceAI	Max SpliceScore	SNPEff	Homer
chr5:150446963:C:T:rs4841	5	ENST00000519690.1	RPS14		NA	NA	SpliceAI=T RPS14 0.21 0.0		0.21	splice_regic exon (NM_005617, exon 3 of 5)
chr5:150447535:G:A:rs3797622	5	ENST00000519690.1	RPS14		NA	NA	SpliceAI=A RPS14 0.00 0.0		0.03	intron_varic intron (NM_001025070, intron 2 of 4)
chr7:142065610:T:C:rs62477626	7	ENST00000475668.6	MGAM		ENSP00000417515.2	2753	SpliceAI=C MGAM 0.03 0.0		0.08	synonymou intron (NM_004668, intron 38 of 47)
chr7:142065610:T:C:rs62477626	7	ENST00000549489.6	MGAM		ENSP00000447378.2	1857	SpliceAI=C MGAM 0.03 0.0		0.08	synonymou intron (NM_004668, intron 38 of 47)
chr12:112919388:G:A:rs10774671	12	ENST00000202917.1	OAS1		ENSP00000202917.5	400	SpliceAI=A OAS1 0.10 0.8		0.89	splice_accce intron (NM_001320151, intron 5 of 5)
chr12:112919388:G:A:rs10774671	12	ENST00000553152.1	OAS1		ENSP00000449053.1	130	SpliceAI=A OAS1 0.10 0.8		0.89	splice_accce intron (NM_001320151, intron 5 of 5)
chr12:112919388:G:A:rs10774671	12	ENST00000452357.6	OAS1		ENSP00000415721.2	364	SpliceAI=A OAS1 0.10 0.8		0.89	splice_accce intron (NM_001320151, intron 5 of 5)
chr12:108619543:A:G:rs73191241	12	ENST00000388962.4	SELPLG		ENSP00000373614.3	402	.			downstrear Intergenic
chr12:108619543:A:G:rs73191241	12	ENST00000550948.1	SELPLG		ENSP00000447752.1	412	.			downstrear Intergenic
chr12:108619543:A:G:rs73191241	12	ENST00000388962.4	SELPLG		ENSP00000373614.3	402	.			downstrear Intergenic

We chose to identify isoform ratio QTLs (irQTLs) primarily for their ability to detect regulatory variation affecting full-length splice isoforms and for their strong interpretability in relation to phenotypic traits. We added the following text on page 18, line 405:

“The isoform ratio method¹⁸ captures the relative expression of each isoform within a gene, allowing the detection of regulatory variation that affects full-length splice isoforms. It is particularly useful for identifying biologically interpretable changes, such as isoform switches that impact coding potential, untranslated regions, or transcript stability. Compared to exon-level QTLs,²¹ which detect local events like exon skipping, and junction-based LeafCutter QTLs,^{16,17} which are optimized for discovering novel or cryptic intron excision, the isoform ratio method provides a clearer view of transcript-level consequences (**Supplemental Table 16**). However, it relies on comprehensive and accurate transcript annotations, potentially missing events in poorly annotated genes. It also has lower resolution to identify specific splice sites or exons, may overlook subtle or complex splicing changes. Despite these limitations, its high biological interpretability and relevance to phenotype-driven studies make it a valuable complement to other QTL approaches.”

Supplemental Table 16. Compare methods: isoform ratio, exon-level QTL, and LeafCutter QTL

Method	Splicing resolution	Annotation dependency	Biological interpretability	Appropriate for
--------	---------------------	-----------------------	-----------------------------	-----------------

Isoform ratio QTL	Low (full transcript)	High	High (e.g., protein isoforms)	Transcript switches
Exon-level QTL	Medium (individual exons)	Medium	Medium	Exon skipping/inclusion
LeafCutter QTL	High (intron junctions)	Low	Low-to-medium	Novel or cryptic splice

2. There is an existing study that performed "sQTL" mapping on the Framingham Heart Study dataset: (Nat Genet 2015; doi: 10.1038/ng.3220). Zhang, X., Joehanes, R., Chen, B. et al. Identification of common genetic variants controlling transcript isoform variation in human whole blood. Nat Genet 47, 345–352 (2015). <https://doi.org/10.1038/ng.3220>

It is strange that this study was not cited or discussed in the present study, and that makes it difficult to assess the novelty of the present study. From a cursory examination, they appear similar.

Reply: We appreciate the opportunity to address this important point. Respectfully, we would like to clarify that our study is not merely a secondary analysis of the 2015 data, as suggested. Rather, it represents a substantially expanded investigation of completely original data.

Below, we highlight several key distinctions between the current work and our earlier publication (**Supplemental Table 1**).

Supplemental Table 1. Comparison of major characteristics between the 2015 paper and the present study (we made the font small to fit in the page).

Characteristics	2015 study	Current study
Study population	Framingham Heart Study	Framingham Heart Study
Discovery n	n=5257	n=2662
SNP identification	Imputation based on SNP array data aligned to hg19	SNPs from whole-genome sequencing aligned to hg38
Number of SNPs	9,883,769	12,887,093
RNA platform	Affymetrix GeneChip Human Exon 1.0 ST aligned to hg19	Whole transcriptome sequencing aligned to hg38
Cis-window definition	± 50 kb from gene boundary	± 1Mb from gene transcription start site (TSS)
Trans-QTL	Not studied	Beyond ± 1Mb from TSS or on the different chromosome
Uncommon/rare QTLs	No	MAF > 0.003 for cis-irQTL
QTL identification method	SNPs associated with exon expression	SNPs associated with isoform ratio
Replication	Findings from the 2015 study were not tested for replication in an independent dataset. The study did replicate some previously reported sQTL-gene pairs from Battle et al. (2014), which used RNA-seq and SNP imputation in 922 individuals.	Internal replication with additional 1094 samples
		External replication in Women's Health Initiative (n=2005) and Jackson Heart Study (n=1010)
Investigation of relationships between irQTLs, alternative splicing, and diseases/traits	No	Provide three examples to investigate the relationships.

In the revised Introduction on page 4, line 86, we added the following:

“Comprehensive analyses of SNP associations with isoform variation in whole blood and their relevance to human health remain limited. In prior work, we identified exon-level expression variants in 2,650 genes using imputed SNPs and exon array data from 5,257 Framingham Heart Study (FHS) participants.²¹ Here, we present a new study leveraging WGS and RNA-seq data from 3,716 FHS participants (2,622 for discovery, 1,094 for internal replication) to map isoform ratio QTLs (irQTLs) as described originally by Yamaguchi et al. (2022)¹⁸ (**Figure 1, Supplemental Table 1**).”

On page 9, line 207, we added a comparison of results from 2015 study and the present study:

“A prior splicing QTL study in FHS identified exon-level QTLs in 2,650 genes using imputed SNPs and exon array data.²¹ That study also reported the most significant splicing QTL within 50 kb of each gene. Only 458 (17.3%) of gene–SNP pairs from the earlier study showed $P < 1 \times 10^{-4}$ in our discovery sample (**Supplemental Table 7**). This modest replication rate likely reflects differences in technology (imputed SNPs vs. WGS; exon arrays vs. RNA-seq) and methodology (SNPs associated with exon expression vs. isoform ratio changes) between the earlier and current studies.”

We added the following text in revised Discussion, page 20, from line 455:

“Our study confirmed a subset of previously reported splicing QTL–gene pairs from FHS,²¹ identifying concordant associations for 17.3% of the 2,650 pairs tested ($P < 1 \times 10^{-4}$). The limited overlap may reflect differences in study design and methodology, including genetic and transcriptomic platforms (imputed SNPs vs. WGS; exon arrays vs. RNA-seq) and the expression features tested (exon-level expression vs. isoform ratios). Some previously reported associations may also represent false positives, highlighting the importance of independent validation with high-resolution data.”

Please also see our revised Results for details (from page 4 to page 10).

3. There is heavy emphasis on the number of variant-phenotype pairs with significant association. The number of such associations is nearly meaningless, since it is well understood that in xQTL studies such as this, the vast majority of variants in the associated pairs are not involved in regulation but rather are in LD with the causal variants. Similarly, the reported percentages of such pairs that were validated by the replication analysis is not meaningful, because even in different cohorts with different ancestries, variants in close proximity are likely to have some level of LD in both cohorts. Therefore, finding that high percentages of variant-phenotype pairs replicate does not indicate how many of the pairs actually have a real causal or functional relationship.

Reply: Thank you for the insightful comment. In the revised manuscript, we now report discovery and replication results, all based on sentinel QTLs.

In the revised Abstract, we revised the text as follows on page 2, line 41:

“In the Framingham Heart Study (FHS discovery set, $n = 2,622$), we identified over 1.1 million *cis*-irQTLs ($MAF \geq 0.01$, within ± 1 Mb of 10,883 isoform transcripts, $P < 5 \times 10^{-8}$) across 4,971 genes. The replication rate ($P < 1 \times 10^{-4}$) for 11,425 most significant (i.e., sentinel) *cis*-irQTLs per isoform was 77% in a separate FHS sample ($n = 1,094$) and 72% in the Women’s Health Initiative (WHI external replication set; $n = 2,005$). Notably, 20% of these *cis*-irQTLs showed no significant association with overall gene expression,

indicating isoform-specific effects. These variants were significantly enriched ($P < 1 \times 10^{-10}$) at splice donor/acceptor sites and GWAS loci. We also identified 1,870 sentinel *trans*-irQTLs ($P < 1.5 \times 10^{-13}$; $MAF \geq 0.01$), located beyond 1 Mb or on different chromosomes, associated with 1,084 isoforms across 590 genes, with similar replication rates. Additionally, we identified 2,327 rare *cis*-irQTLs ($0.003 < MAF < 0.01$, $P < 5 \times 10^{-8}$) for 2,467 isoforms of 1,428 genes, 22% of which replicated internally and 41% replicated externally in WHI.”

In revised Results, page 5, line 113,

“*Cis*-irQTL variants with minor allele frequency ($MAF \geq 0.01$)

We identified over 1.1 million irQTLs associated with 10,883 isoform transcripts, representing more than 3.5 million significant *cis*-irQTL-isoform pairs ($P < 5 \times 10^{-8}$) across 4,971 genes. Of these, we prioritized 11,425 sentinel variants forming 14,056 lead pairs with the same isoforms (**Supplemental Table 3**). A *cis*-irQTL is defined as a variant located within 1 Mb of an isoform’s transcription start site (TSS), and a sentinel *cis*-irQTL is the variant most strongly (i.e., lowest p-value) associated with an isoform. An isoform may be associated with multiple sentinel *cis*-irQTLs if they are in mutual perfect linkage disequilibrium ($LD r^2 = 1$). The top *cis*-irQTL-Gene pairs are listed in Table 2.”

In revised Results, page 6, line 142:

“*Trans*-irQTLs with $MAF \geq 0.01$

A *trans*-irQTL was defined as being located more than 1 Mb away from the TSS of an isoform or on a separate chromosome. Using a significance threshold of $P < 1.5 \times 10^{-13}$, we identified over 130,000 *trans*-irQTLs associated with 1,084 isoforms from 590 genes. For each isoform, we selected the most significant *trans*-irQTL per chromosomes (22 autosomes and X chromosome), yielding 1,870 sentinel *trans*-irQTLs in the discovery sample (**Supplemental Table 4**).”

In revised Results, page 7, line 162:

“Rare *cis*-irQTLs

We assessed *cis*-irQTL variants ($0.003 < MAF < 0.01$) in the discovery sample. This analysis identified 2,327 rare *cis*-irQTL variants associated with 2,467 isoforms ($P < 5 \times 10^{-8}$), yielding 3,102 significant *cis*-irQTL-isoform pairs across 1,428 genes (**Supplemental Table 5**).”

In revised Results, page 8, line 175:

“Internal and external replication of irQTLs

Replication was performed for irQTL–isoform pairs associated with sentinel variants. An irQTL-isoform pair from a sentinel variant was considered replicated if it was found in the replication sample with a $P < 1 \times 10^{-4}$ and showed the same direction of association as in discovery.”

Please also see detailed results from page 4 to page 10 in the revised manuscript.

Other comments:

1. Line 128: "They found that expression levels of nearly all expressed genes (eGenes) are subject to genetic regulation." The cited GTEx study defines 'eGenes' as "genes with an eQTL", not "expressed genes".

Reply: Thank you for pointing this out. In the revised manuscript, we have modified the sentence as follows (on page 3, line 78):

"These studies established the widespread influence of genetic variation on transcriptional regulation, including splice variation.²²"

2. The abstract mentions "allelic variation" twice in reference to a specific analysis, but doesn't define it, and the term is not used again in the text, so I can't tell if the analysis is actually presented. Perhaps it is just referring to the genotypic variation on which the QTLs are determined, but then this sentence: "We identified over 3.5 million cis-sQTL-isoform pairs ($p < 5e-8$), comprising 1,176,624 cis-sQTL variants and 10,883 isoform transcripts from 4,971 sGenes, with significant change in isoform-to-gene ratio due to allelic variation." would certainly be false, as the end of the sentence asserts a causal relationship, which is not the case for the vast majority of those 3.5 million cis-sQTL-isoform pairs.

Reply: Thank you for this comment. In the revised Abstract, the revised text is below, on page 2, line 46:

"Notably, 20% of these cis-irQTLs showed no significant association with overall gene expression, indicating isoform-specific regulation."

The Reviewer is also correct for pointing out the confusion about causal relationships. A causal relationship is not appropriate here to describe *cis-irQTL-isoform pairs*. We have replaced "due to" with "associated with" in the Abstract. We have also updated our results to focus on sentinel irQTLs rather than all significant irQTLs, rendering the original sentence no longer relevant. Nevertheless, we have used the phrase 'associated with' throughout the revised manuscript where appropriate and have revised instances of 'due to' to more appropriate alternatives.

Reviewer #3 (Remarks to the Author):

This manuscript identifies splice QTL in blood to better understand the genetic influences on alternative splicing and alternative splicing as a potential molecular mechanism for how GWAS SNPs influence disease. The authors use isoform ratios as their quantitative measure of alternative splicing and include both a within sample validation (i.e., split the original cohort into a discovery and validation set of samples) and a cross-population validation (i.e., samples from a completely independent cohort with many differences in sample demographics and in RNA isolation and processing). This is a well-thought-out study with a tremendous amount of data and impactful results. In general, major issues below are focused on additional information to include in the discussion. As written, the discussion sounds more like a summary of the results than a discussion of implications and relevant assumptions.

Major Issues

1. Authors state on line 141 of the introduction that one of the aims of this study was to ‘create a comprehensive sQTL resource’, but it was not clearly stated how the research public could gain access to the sQTL resource.

Reply: We uploaded all of our results to dbGaP, where they are accessible by the broad scientific community.

2. One point that was not included in the discussion is potential differences in sQTL analyses that use isoform ratios vs eQTL analyses done with expression levels of individual isoforms in both methodology and in interpretation.

Reply: We thank the reviewer for raising this point. We have added the following text in the revised Discussion, on page 18, line 405:

“The isoform ratio method¹⁸ captures the relative expression of each isoform within a gene, allowing the detection of regulatory variation that affects full-length splice isoforms. It is particularly useful for identifying biologically interpretable changes, such as isoform switches that impact coding potential, untranslated regions, or transcript stability. Compared to exon-level QTLs,²¹ which detect local events like exon skipping, and junction-based LeafCutter QTLs,^{16,17} which are optimized for discovering novel or cryptic intron excision, the isoform ratio method provides a clearer view of transcript-level consequences (**Supplemental Table 16**). However, it relies on comprehensive and accurate transcript annotations, potentially missing events in poorly annotated genes. It also has lower resolution to identify specific splice sites or exons, may overlook subtle or complex splicing changes. Additionally, the isoform ratio method introduces interdependency among isoforms of the same gene, as their proportions must sum to 1. Consequently, an increase in the expression ratio of one isoform may correspond to a decrease in one or more others. Despite these limitations, its high biological interpretability and relevance to phenotype-driven studies make it a valuable complement to other QTL approaches..”

Supplemental Table 16. Compare methods: isoform ratio, exon-level QTL, and LeafCutter QTL

Method	Splicing resolution	Annotation dependency	Biological interpretability	Appropriate for
--------	---------------------	-----------------------	-----------------------------	-----------------

Isoform ratio QTL	Low (full transcript)	High	High (e.g., protein isoforms)	Transcript switches
Exon-level QTL	Medium (individual exons)	Medium	Medium	Exon skipping/inclusion
LeafCutter QTL	High (intron junctions)	Low	Low-to-medium	Novel or cryptic splice

3. No discussion was included related to the dependency between sQTL tests of different isoforms of the same gene. Throughout, it was shown that the number of significant isoforms is approximately twice the number of significant genes.

Reply: Thank you for your comments. In the revised Discussion, we added the following to address the dependency between irQTL tests of different isoforms of the same gene, on page 18, from line 415 (also see our reply to this Reviewer’s Comment 2):

“Additionally, the isoform ratio method introduces interdependency among isoforms of the same gene, as their proportions must sum to 1. Consequently, an increase in the expression ratio of one isoform may correspond to a decrease in one or more others. Despite these limitations, its high biological interpretability and relevance to phenotype-driven studies make it a valuable complement to other QTL approaches.”

As noted by the reviewer, the number of significant isoforms is approximately twice the number of significant genes overall (e.g., 10,883 isoforms across 4,971 genes). Upon reviewing the data, we found that this appears to be coincidental. We summarized the significant isoforms (n=10,883) per gene for n=4,971 genes. Although each gene may have multiple isoforms, only one or a few may be identified as significant. For example, among the 4,971 genes, 2,087 have exactly one significant isoform, while in one case, a single gene has 19 significant isoforms.

#significant isoforms	# Gene	Total
1	2087	2087
2	1529	3058
3	647	1941
4	310	1240
5	162	810
6	101	606
7	55	385
8	36	288
9	16	144
10	11	110
11	6	66
12	4	48
13	4	52
14	1	14
15	1	15
19	1	19
Total	4971	10883

4. There was little discussion related to the potential impact of dramatic differences in the FHS and JHS populations including race, treatment for hypertension, high lipid levels, and diabetes.

Reply: Thank you for the insightful comments. The low replication rate in JHS likely reflects differences in RNA source (whole blood in FHS vs. PBMCs in JHS). To better address this concern, we incorporated an additional replication cohort, the Women's Health Initiative (WHI), which includes WGS and RNA-seq data derived from whole blood. We conducted analyses in the full WHI sample as well as separately in White American and African American subgroups. This allowed us to investigate whether the lower replication rate observed in JHS may be more attributable to differences in RNA source, race, or clinical characteristics. Relevant results and discussion have been included in the revised manuscript.

In the revised Abstract, we revised the text as follows on page 2, line 41:

“In the Framingham Heart Study (FHS discovery set, $n = 2,622$), we identified over 1.1 million *cis*-irQTLs ($MAF \geq 0.01$, within ± 1 Mb of 10,883 isoform transcripts, $P < 5 \times 10^{-8}$) across 4,971 genes. The replication rate ($P < 1 \times 10^{-4}$) for 11,425 most significant (i.e., sentinel) *cis*-irQTLs per isoform was 77% in a separate FHS sample ($n = 1,094$) and 72% in the Women's Health Initiative (WHI external replication set; $n = 2,005$). Notably, 20% of these *cis*-irQTLs showed no significant association with overall gene expression, indicating isoform-specific effects. These variants were significantly enriched ($P < 1 \times 10^{-10}$) at splice donor/acceptor sites and GWAS loci. We also identified 1,870 sentinel *trans*-irQTLs ($P < 1.5 \times 10^{-13}$; $MAF \geq 0.01$), located beyond 1 Mb or on different chromosomes, associated with 1,084 isoforms across 590 genes, with similar replication rates. Additionally, we identified 2,327 rare *cis*-irQTLs ($0.003 < MAF < 0.01$, $P < 5 \times 10^{-8}$) for 2,467 isoforms of 1,428 genes, 22% of which replicated internally and 41% replicated externally in WHI.”

In the revised Discussion, we added the following, on page 19, line 435:

“In addition to FHS, we analyzed community-based cohorts, JHS²³ and WHI²⁴, to enable independent replication and enhance the generalizability of our findings across diverse populations. We observed a 72% replication rate in WHI, supporting the robustness of our results. In contrast, replication in JHS was lower (32%), likely reflecting both technical and biological differences. Notably, JHS used RNA-seq from PBMCs, while FHS and WHI used whole blood. To further assess the impact of tissue source and ancestry, we stratified WHI by ancestry. Replication was highest in the full WHI cohort and lower in ancestry-specific subsets, partly reflecting reduced sample size (**Supplemental Figure 2**). Despite similar sample sizes, replication in JHS African American participants was lower than in WHI African American participants, suggesting that tissue source may have a greater effect on replication than ancestry. While clinical characteristics also varied across cohorts (**Table 1**, **Supplemental Table 2**), their influence on splicing could not be directly assessed. These results underscore the importance of considering tissue type, ancestry, and sample size in cross-cohort transcriptomic analyses.”

Minor Comments

1. A table that compared the source and processing of the RNA samples between the JHS and FHS would help clarify differences and similarities between studies. The description of the processing in the methods section was not easily comparable between the two studies.

Reply: Thanks for the suggestion. We added the table as Supplemental Table 7, see below:

Supplemental Table 7. Comparison of the source and processing of RNA samples between the FHS and JHS cohorts

	FHS	JHS
Sample size	2,622	1,010
Self-reported race	>95 White American participants	African American participants
RNA source	Whole blood	Peripheral blood mononuclear cells
Sequencing center	University of Washington Northwest Genomics Center (NWGC)	University of Washington Northwest Genomics Center (NWGC)
Processing	RSEM (v1.3.1)	RSEM (v1.3.1)
Sequence alignment	GRCh38 with reference transcriptome GENCODE 30 using STAR (v2.7.0a)	GRCh38 with reference transcriptome GENCODE r30 using STAR (v.2.6.1d)
Isoform quantification	Fragments Per Kilobase of transcript per Million mapped reads (FPKM)	Fragments Per Kilobase of transcript per Million mapped reads (FPKM)

Please also see our response to this Reviewer’s major Comment 4.

2. More details are needed about how transcripts are quantitated using STAR. For example, how were multi-mapping reads handled? What the max number of multiple alignments that were reported?

Reply: Thank you for this question. Many of the needed details are now included in the Supplementary Materials, including a list of parameters used when running STAR. The calls made by the STAR program are part of a standardized pipeline in TOPMed, a summary description is available at https://github.com/broadinstitute/gtex-pipeline/blob/master/TOPMed_RNAseq_pipeline.md. The STAR option switch – outFilterMultimapNmax defaults to a value of 20, meaning that reads are only output if they map to 20 or fewer locations.

3. It is curious that more cis pairs validated in FHS validation set than trans pairs. There was no mention of this in the discussion.

Reply: We think that the reviewer meant to ask why more *trans* pairs validated in the FHS replication set than *cis* pairs. **Please see below from the original submission.**

“In the FHS discovery sample, we identified a total of 3,588,153 significant *cis*-sQTL-isoform pairs, comprising 1,176,624 distinct *cis*-sQTL variants and 10,883 distinct isoform transcripts from 4,971 distinct genes (**Table 2**). Of these, 61% of the *cis* pairs were validated in the FHS replication sample. We also identified 328,997 significant *trans*-sQTL-isoform pairs, consisting of 131,804 distinct *trans*-sQTL variants and 1,054 distinct isoform transcripts from 590 distinct genes in the FHS discovery sample. Compared to *cis*-pairs, a larger proportion of *trans*-sQTL-isoform pairs (n=235,807, 72%) were validated in the FHS replication sample (**Table 2**). The sentinel *cis*-sQTL and *trans*-sQTLs identified in FHS were presented in **Supplemental Tables 2 and 3’**.”

We believe this reflects consistent conditions and procedures used to process blood samples, extract RNA, conduct RNA-seq, and analyze the data. In the revised manuscript, we presented sentinel irQTLs, as suggested by the reviewers. We found a consistent trend for sentinel *trans*-irQTLs as we did for all significant pairs in our previous version. However, in the WHI external replication, the replication rate was higher for *cis* than *trans* pairs. In the revised Discussion, on page 20, line 450, we added the following to discuss our findings:

“Using our approach, *cis*-irQTLs showed higher external replication rates than *trans*-irQTLs in WHI (n = 2,005), consistent with published findings,^{58,59} likely reflecting their proximity to target isoforms and involvement in local regulation. Nevertheless, we observed more replicated *trans*-irQTL-isoform pairs than *cis* pairs in the additional FHS sample (n = 1,094), which may be explained by consistent experimental conditions and shared analytical pipelines across FHS discovery and replication sample.”

4. *Line 391 – it isn't clear why there would be more 'distinct sentinel cis-sQTL' than isoforms based on how a sentinel cis-sQTL was defined.*

Reply: More than one *cis*-SNP in perfect linkage disequilibrium (LD, $r^2 = 1$) may be associated with the same isoform, resulting in more sentinel *cis*-irQTLs than isoforms.

We added this missing explanation to the revised Results on page 5, from line 120:

“An isoform may be associated with multiple sentinel *cis*-irQTLs if they are in mutual perfect linkage disequilibrium (LD $r^2 = 1$).”

5. *Table 3 – if all cis-sQTL isoform pairs are replicated in this table, why do some have NA values for the JHS cohort?*

Reply: Table 3 in the previous version of the manuscript is now Table 2. We added WHI as our external replication cohort, and the revised Table 2 no longer contains any 'NA' entries.

6. *Line 418 – The point of this introductory sentence is unclear. How are they judging that an isoform is strongly regulated by genetic variants? The cis-eQTL-Gene is at the gene-level, not the isoform or transcript level?*

Reply: We agree with your comment. The Introduction has been substantially revised, and the sentence in question has been removed as it is no longer relevant.

7. *It would be informative to compare the enrichment of cis-sQTL in mQTL, pQTL, and GWAS to the enrichment of cis-eQTL in these groups.*

Reply: We thank the reviewer for this insightful comment.

We are currently investigating the relationship between pQTLs or mQTLs and irQTLs in two ongoing projects, with separate manuscripts in preparation. Given the ambitious scope of the current study and the intricate nature of irQTL-pQTL and irQTL-mQTL relationships, we have chosen to remove the enrichment analyses from this manuscript. These analyses will be presented in greater depth in forthcoming dedicated papers.

8. Line 516 – referring to wrong figure.

Reply: In the revised manuscript, we carefully reviewed the cited Figure numbers and corrected all typos.

9. It would be helpful to be consistent on which strand you are using to derive reported genotype, e.g., line 505 refers to T/T, T/C, and C/C whereas figure refers to A/A, A/G, and G/G

Reply: Thank you for the suggestion. We have revised the manuscript to consistently use the same strand for reporting genotypes across the text and figures.

In the revised manuscript, page 14, from line 315, we added the following clarification:

“As *ULK3* is transcribed from the negative strand, we converted the sequence surrounding rs12898397 to its complementary form (T>C → A>G) while maintaining the 5' to 3' orientation for downstream analyses.”

In the revised Figure 5, we revised the legend (page 37) as follows:

“**Figure 5A.** *ULK3* (unc-51 like kinase 3) lies on the negative strand of chromosome 15 (hg38) and contains 16 exons. The sentinel irQTL rs12898397 (T>C) is located in exon 14. To preserve transcriptional orientation (5'→3'), we converted the positive-strand sequence around rs12898397 to its complementary negative-strand sequence (e.g., T>C → A>G.”

10. Line 532 – refers to the wrong figure panel

Reply: In the revised manuscript, we corrected the typo.

11. Lines 709-711 – the last sentence of the paragraph is unclear.

Reply: In the revised manuscript, we added analysis of rare variants ($0.003 < \text{MAF} < 0.01$). The last sentence is no longer relevant.

12. Perhaps an alternative type of graphic would make Figure 2 more impactful, e.g., violin plots by quantile. Too many points make the denser areas hard to interpret.

Reply: In our revised manuscript, we updated Figure 2 with its legend as shown below.

Figure 2. Replication of sentinel *cis*-irQTLs in relation to their associated isoforms. We identified 11,425 common variants ($MAF \geq 0.01$) as sentinel *cis*-irQTLs for 10,883 isoforms across 4,971 genes in the FHS discovery sample ($n = 2,622$). Internal replication was performed in an independent FHS sample ($n = 1,094$), and external replication in the WHI cohort ($n = 2,005$). A sentinel *cis*-irQTL–isoform ratio association was considered replicated if $P < 1 \times 10^{-4}$ and the effect direction matched the discovery. **(A)** Replication rates increased with R^2 , the proportion of isoform ratio variance explained by the irQTL. Average replication rates were 76.7% in FHS and 72.4% in WHI. **(B)** Comparison of effect sizes between the FHS discovery and WHI replication samples showed strong consistency, with 99.8% of *cis*-irQTL-isoform pairs sharing the same direction of association.

13. The take home message of figure 3B is not clear. Too many overlapping points make it hard to decipher.

Reply: We revised Figure 3B (see below) and revised the legend to provide a clearer take-home message.

Figure 3B. Comparison of association strength (R^2) for *cis*-eQTLs and *cis*-irQTLs across 4,971 genes with detected irQTLs. Dashed lines define three SNP groups: 1,946 with stronger irQTL signals (cyan), 2,716 with overlapping irQTL and eQTL signals (middle), and the remainder showing stronger eQTL signals (red).

14. Y-axis labels on Figure 5D would be helpful.

Reply: We added the Y-axis label in the revised Figure (Now Figure 5B).

15. Labeling the location of the SNP on Figure 4A would be helpful.

Reply: We added the location of the SNP on Figure 4A. We also modified the legend for 4A.

Figure 4A. *OAS1* is located on the long arm of chromosome 12 on the positive strand in the hg38 human genome build. Its pre-mRNA undergoes editing and alternative splicing. The variant rs10774671 (G>A) lies at the splice acceptor site of intron 5.